



Atmospheric
Measurement
Techniques

# Product ion distributions using $H_3O^+$ proton-transfer-reaction time-of-flight mass spectrometry (PTR-ToF-MS): mechanisms, transmission effects, and instrument-to-instrument variability

Michael F. Link[1], Megan S. Claflin[2], Christina E. Cecelski[1], Ayomide A. Akande[3], Delaney Kilgour[4], Paul A. Heine[3], Matthew Coggon[5], Chelsea E. Stockwell[5], Andrew Jensen[6,a], Jie Yu[7], Han N. Huynh[7,b], Jenna C. Ditto[7,c], Carsten Warneke[5], William Dresser[6], Keighan Gemmell[3], Spiro Jorga[7,d], Rileigh L. Robertson[1,e], Joost de Gouw[6], Timothy Bertram[4], Jonathan P. D. Abbatt[7], Nadine Borduas-Dedekind[3], and Dustin Poppendieck[1]

[1]National Institute of Standards and Technology, Gaithersburg, 20899, USA
[2]Aerodyne Research Inc., Billerica, 01821, USA
[3]Department of Chemistry, University of British Columbia, Vancouver, V6T 1Z1, Canada
[4]Department of Chemistry, University of Wisconsin–Madison, Madison, 53706, USA
[5]National Oceanic and Atmospheric Administration, Boulder, 80305, USA
[6]Department of Chemistry, University of Colorado Boulder, Boulder, 80309, USA
[7]Department of Chemistry, University of Toronto, Toronto, M5S 3H6, Canada
[a]now at: Department of Chemistry, University of Michigan, Ann Arbor, 48109, USA
[b]now at: National Oceanic and Atmospheric Administration, Boulder, 80305, USA
[c]now at: Energy, Environmental & Chemical Engineering, Washington University in St. Louis, St. Louis, 63130, USA
[d]now at: Tofwerk, Thun, 3645, Switzerland
[e]now at: Department of Mechanical Engineering, University of Colorado Boulder, Boulder, 80309, USA

**Correspondence:** Michael F. Link (michael.f.link@nist.gov)

**Abstract.** Proton-transfer-reaction mass spectrometry (PTR-MS) using hydronium ion ($H_3O^+$) ionization is widely used for the measurement of volatile organic compounds (VOCs) both indoors and outdoors. $H_3O^+$ ionization, as well as the associated chemistry in an ion–molecule reactor, is known to generate product ion distributions (PIDs) that include other product ions besides the proton-transfer product. We present a method, using gas-chromatography pre-separation, for quantifying PIDs from PTR-MS measurements of nearly 100 VOCs of different functional types including alcohols, ketones, aldehydes, acids, aromatics, organohalides, and alkenes. We characterize instrument configuration effects on PIDs and find that reactor reduced electric field strength ($E/N$), ion optic voltage gradients, and quadrupole settings have the strongest impact on measured PIDs. Through an interlaboratory comparison of PIDs measured from calibration cylinders, we characterized the variability of PID pro-

duction from the same model of PTR-MS across seven participating laboratories. Product ion variability was generally smaller (e.g., < 20 %) for ions with larger contributions to the PIDs (e.g., > 0.30) but less predictable for product ions formed through $O_2^+$ and $NO^+$ reactions. We present a publicly available library of $H_3O^+$ PTR-MS PIDs that will be updated periodically with user-provided data for the continued investigation into instrument-to-instrument variability of PIDs.

## 1 Introduction

Measurements of volatile organic compounds (VOCs) using hydronium ion ($H_3O^+$) proton-transfer-reaction mass spectrometry (PTR-MS) have become ubiquitous in a variety of applications in the past 25 years (Yuan et al., 2017; Sekimoto

and Koss, 2021). PTR-MS can measure many VOCs simultaneously with fast ($> 1\,\mathrm{Hz}$) time resolution and low detection limits (e.g., $< 1\,\mathrm{nmol\,mol^{-1}}$) and is selective towards VOCs that have a proton affinity greater than water (e.g., ketones, aldehydes, nitriles) (De Gouw et al., 2003). However, in the absence of sample pre-separation, isobaric (i.e., same mass-to-charge ratio, $m/q$) interferences are known to pose challenges to VOC identification and quantification (Coggon et al., 2024; Kilgour et al., 2024; Ditto et al., 2025). Since the early development of PTR-MS, studies have shown that unintended product ions can complicate mass spectra (Warneke et al., 2003; De Gouw and Warneke, 2007), but more recent studies have highlighted ion interferences in measurements of urban air plumes (Coggon et al., 2024) and indoor air (Ernle et al., 2023; Ditto et al., 2025), where interferences are pronounced because VOC concentrations are high and emission sources are diverse. As PTR-MS technology continues to improve through the development of new sample introduction methods, ionization technologies (Krechmer et al., 2018; Breitenlechner et al., 2017; Reinecke et al., 2023), and enhanced mass resolution through the use of time-of-flight mass analyzers, this method will continue to be utilized in concentrated and chemically diverse sample matrices. The popularity of this measurement technique warrants the creation of standardized methods for measuring and quantifying the effects of unintended, or poorly understood, product ion distributions on PTR-MS mass spectra.

Unintended product ion generation in PTR-MS has been discussed extensively, including in studies highlighting the importance of VOC fragmentation from $H_3O^+$ ionization (e.g., aldehydes, Ernle et al., 2023; peroxides, Li et al., 2022; and monoterpenes, Misztal et al., 2012; Kari et al., 2018; Tani, 2013) and studies using selected-ion flow tube (SIFT) reaction measurements (summarized in a recent review by Hegen et al., 2023) to differentiate interferences from $O_2^+$ and $NO^+$ reagent ion impurities. Pagonis et al. (2019) presented a library of previously reported product ion distributions (PIDs) compiled from measurements of VOCs. However, water cluster contributions to the PIDs were largely not represented in this compilation. The library shows considerable variability in the generation of product ions for a given VOC (e.g., butanal, ethyl acetate), but from the existing data it is not clear if this variability is explained by instrument operating parameters, features of the specific instrument, or methods of quantifying PIDs.

In this study we highlight

1. a gas-chromatographic method for measuring PIDs from the ionization of VOCs using PTR-MS (Sect. 2.2);

2. how instrument configurations can influence PIDs (Sect. 3.1);

3. instrument-to-instrument variability in measured PIDs determined from an interlaboratory comparison (Sect. 3.2);

4. the propensity of different VOC functional types to form complex PIDs that include water clusters (Sect. 3.3);

5. an example of how PIDs can cause ambiguity when identifying ions using a sample of restroom air as a case study (Sect. 3.4);

6. suggestions of how PIDs can be used to aid in identification and quantification of VOCs from PTR-MS mass spectra (Sect. 3.5);

7. a library of $H_3O^+$ PTR-MS PIDs available for community use, to be updated with continued collaborative input, and uncertainty estimates (Sect. 3.6); and

8. recommendations for mitigating and managing unintended product ion generation using PTR-MS (Sect. 3.7).

## 2 Materials and methods

### 2.1 Product ion definitions and formation mechanisms

We use observations from previous studies (Koss et al., 2016; Xu et al., 2022; Pagonis et al., 2019; Hegen et al., 2023; Coggon et al., 2024; Li et al., 2024) to identify the reactions and associated product ions that are likely to be important from $H_3O^+$ (and impurity $NO^+$ and $O_2^+$) ionization of a given VOC. The reaction mechanisms we identify here do not represent an exhaustive accounting of possible product ion formation mechanisms but instead represent mechanisms most likely to generate the product ions observed from our data. VOCs ($M = VOC$) with a proton affinity greater than water ($691\,\mathrm{kJ\,mol^{-1}}$) can undergo a proton-transfer reaction with $H_3O^+$ to form an $H^+$ adduct (labeled as $MH^+$) as described in Reaction (R1).

$$M + H_3O^+ \rightarrow MH^+ + H_2O \qquad (R1)$$

Unique from most previous studies, we quantify the contribution of protonated VOC water clusters (labeled as $[MH \cdot (H_2O)_n]^+$, where $n = 1$ or 2) to the product ion distribution that potentially form from direct association reactions following Reaction (R2) (Li et al., 2024) and/or termolecular association reactions of a protonated VOC with water vapor following Reaction (R3).

$$M + H_3O^+ + B \rightarrow [M \cdot H_3O]^+ + B \qquad (R2)$$

$$MH^+ + H_2O + B \rightarrow [MH \cdot H_2O]^+ + B \qquad (R3)$$

The presence of a collisional body, $B$ ($B = N_2$ or $O_2$), in Reactions (R2) and (R3) implies a pressure dependence (McCrumb and Warneck, 1977; Smith et al., 2020). Direct protonation and water cluster formation can also occur from reaction of VOCs with reagent ion water clusters (De Gouw and

Warneke, 2007).

$$M + (H_2O)_n H_3O^+ \rightarrow [M \cdot H_3O]^+ + (H_2O)_n \qquad (R4)$$

$$M + (H_2O)_n H_3O^+ \rightarrow [M \cdot (H_2O) H_3O]^+ + (H_2O)_{n-1} \quad (R5)$$

However, the addition of the radio frequency (RF)-only quadrupole around the ion–molecule reactor (IMR; in the instruments evaluated in this study) serves to decrease the influence of higher-order water clusters on ionization chemistry (Krechmer et al., 2018). We note that unlike other PTR-MS instruments, the Vocus PTR-ToF-MS instruments featured in this study have been observed to have ionization chemistry that is not appreciably sensitive to sample water vapor concentrations (Krechmer et al., 2018; Li et al., 2024).

Fragmentation of a protonated VOC can occur from the loss of neutral constituents (e.g., $H_2O$, CO, and $C_2H_4O_2$) and/or the dissociation of carbon–carbon bonds (Pagonis et al., 2019). We refer to product ions that result from a fragmentation reaction where water is lost from the protonated VOC, following Reaction (R6), as dehydration products (labeled as $[MH-H_2O]^+$).

$$MH^+ \rightarrow [MH-H_2O]^+ + H_2O \qquad (R6)$$

We highlight the formation of dehydration products because this type of fragment ion contributed the most to a PID of oxygenated VOCs from our dataset. Because other fragmentation product ions could form through a variety of mechanisms (including from reactions with $NO^+$ and $O_2^+$), we label other fragmentation product ions as $F_n$, where $n = 1, 2, 3$, etc.

We highlight two other reaction mechanisms, charge transfer and hydride transfer, that are responsible for generating product ions that often appear in PTR-MS mass spectra. Charge transfer reactions, between a VOC and impurity reagent ions like $O_2^+$ and $NO^+$, can form product ions (labeled as $M^+$) that appear in the mass spectrum as ionized VOCs with no changes to elemental composition (Reaction 7).

$$M + O_2^+/NO^+ \rightarrow M^+ + O_2/NO \qquad (R7)$$

Reactions with $NO^+$ can also ionize VOCs via hydride transfer (labeled as $[M-H]^+$; Reaction R8) (Koss et al., 2016; Spanel and Smith, 1997).

$$M + NO^+ \rightarrow [M-H]^+ + HNO \qquad (R8)$$

We note that Hegen et al. (2023) recently proposed that product ions appearing in mass spectra as hydride transfer products from reactions with $O_2^+$ may actually be charge transfer products that lose a neutral hydrogen atom. For the purposes of this study we classify any product ion that appears in the mass spectrum with the formula $[M-H]^+$ as a hydride transfer product. $NO^+$ and $O_2^+$ ion chemistry can also produce additional product ions through other mechanisms (e.g., hydroxide transfer) not discussed here but which are summarized in

Hegen et al. (2023). We note that in the Vocus instruments used in this study the ratio of $NO^+$ and $O_2^+$ to $H_3O^+$ generated reagent ions cannot be precisely controlled prior to ionization of VOCs in the IMR.

We use the above mechanisms for defining the main product ions considered in our analysis and the rules for determining their location in the mass spectrum, relative to the molecular weight (MW) of the VOC, when calculating PIDs (Table 1).

## 2.2 Method of quantifying PIDs from GC-PTR-ToF-MS measurements

### 2.2.1 Measurement of PIDs using gas-chromatography proton-transfer-reaction time-of-flight mass spectrometry (GC-PTR-ToF-MS)

We used gas-chromatography (GC) pre-separation as a technique for isolating VOCs from multi-component standards before their measurement by the proton-transfer-reaction time-of-flight mass spectrometer (PTR-ToF-MS) to reduce the influence of PIDs from other interfering VOCs. A step-by-step procedure for reproducing this method is presented in the Supplement. PIDs were measured by our group and collaborating lab partners by first separating target analytes from a VOC mixture using GC and then measuring the product ions from $H_3O^+$ ionization (including ionization by impurity reagent ions $O_2^+$ and $NO^+$) of the separated VOC using time-of-flight mass spectrometry (Claflin et al., 2021; Vermeuel et al., 2023). We discuss the details of individual labs' instrument operation below in Sect. 2.5. Most of the PIDs for the individual VOCs we report here, including measurements from instruments participating in the interlaboratory comparison, were measured from calibration cylinders containing multiple VOCs, while Lab 1 measured some PIDs

**Table 1.** Definitions of product ions that occur in PTR-MS mass spectra.

| Product ion identity | Product ion label | Mass-to-charge ratio (Th)* |
|---|---|---|
| $H^+$ adduct | $MH^+$ | $MW + 1.007$ |
| Single water cluster | $[MH \cdot H_2O]^+$ | $MW + 19.018$ |
| Double water cluster | $[MH \cdot (H_2O)_2]^+$ | $MW + 37.028$ |
| Charge transfer | $M^+$ | $MW - 0.001$ |
| Hydride transfer | $[M-H]^+$ | $MW - 1.007$ |
| Dehydration | $[MH-H_2O]^+$ | $MW - 18.011$ |
| Fragment | $F_n$, $n = 1$ through 5 | variable |
| Other | other | variable |

* We express mass-to-charge ratio ($m/q$) in units of thomson (Th); 1 Th $= 1.0364 \times 10^{-8}$ kg C$^{-1}$.
For our analyses we limited the total number of fragment ions that contribute to a PID to five. Most VOCs did not generate more than two fragment ions. Some VOCs (e.g., aromatics generating $C_6H_7O^+$) generated product ions that were consistently observed, but we could not easily explain how they formed, and so we classify these few ions as "other".

by sampling an airstream of evaporated liquid VOC solution. All calibration gas cylinders were less than 2 years old. VOC sources are listed in the $H_3O^+$ PID library included here as a supplemental document but also available online (https://doi.org/10.18434/mds2-3582; Link, 2024). We found that PIDs were difficult to quantify from VOCs measured from ambient air samples due to the potential influence of coeluting VOCs on the determination of the background subtracted mass spectra. However, because of a lack of calibration standards, we included PIDs measured from ambient samples for ethanol and $\alpha$-pinene measured by Lab 6 as well as a monoterpene acetate ester measured by Lab 1. Sample concentrations varied depending on cylinder or liquid solution concentrations, but target VOC concentrations were always less than 10 nmol mol$^{-1}$.

All the data presented in this paper were collected on the Lab 1 PTR-ToF-MS, unless otherwise noted such as in Sect. 3.2, where we compare PIDs measured from different instruments. We differentiate between the seven different laboratories that contributed data by labeling the data as coming from Labs 1 through 7 (e.g., Lab 1). Each instrument used a GC for pre-separation of VOC mixtures and a Vocus time-of-flight mass spectrometer with $H_3O^+$ ionization for subsequent measurement of PIDs. In principle, the chemistry discussed here applies to all PTR-MS instruments that use $H_3O^+$ chemical ionization, but differences in ionization technology, ion transfer optics, and mass analyzers between instruments may have instrument-specific effects on PID measurements. Limited evidence suggests that the PIDs resulting from fragmentation in the Vocus PTR-ToF-MS, as used in this study, and a PTR-MS using a drift tube (instead of an ion–molecule reactor) are comparable (Krechmer et al., 2018), but we limit the implications of our measurements to Vocus PTR-ToF-MS (Tofwerk) instruments until future studies comparing PIDs from different PTR-MS instruments can be performed. The mass spectrometer for Lab 5 used a modified version of the Vocus ionization source (Gkatzelis et al., 2024; Coggon et al., 2024), and the mass spectrometer for Labs 4 and 5 had a lower mass resolution compared to the other instruments (approximately 4000 versus 10 000 full width at half maximum, respectively). Lab 5 also used a custom-built GC, whereas all the other instruments used a commercially available GC (Aerodyne Research Inc.). Because the principle of operation was similar for all instruments, we describe in more detail below the operation of the Lab 1 instrument. Operating details for each of the instruments in the interlaboratory comparison are included in the $H_3O^+$ PID library (also outlined in Table 2).

We describe the GC sampling method used for Lab 1 below but note that operational differences may have been utilized for the different labs represented in the interlaboratory comparison (e.g., temperatures and makeup flow rates). Analytes from multi-component VOC samples were first collected using thermal desorption preconcentration ahead of the chromatographic separation be-

fore ionization by the PTR-ToF-MS. For the laboratories that utilized the commercial GC systems, sample air was passed at a rate of 100 cm$^3$ min$^{-1}$ over a multibed sorbent tube (containing Tenax TA, graphitized carbon, and Carboxen 1000) where VOCs were collected for 10 min. The VOCs were then desorbed from the sorbent tube and collected onto a second preconcentration stage, a focusing trap. VOCs were then rapidly desorbed from the focusing trap and injected on a mid-polarity column (Restek MXT-624, 30 m $\times$ 0.25 mm $\times$ 1.4 µm). VOCs were separated with a helium carrier gas flow of 2 cm$^3$ min$^{-1}$ during the temperature-programmed chromatographic separation. Analytes eluting from the column passed through a transfer line, were heated to 100 °C, and were combined with 150 cm$^3$ min$^{-1}$ of ultrapure zero air before being sampled by the PTR-ToF-MS. Chromatograms were collected over 10 min. Versions of the GC system used in this study are described in detail elsewhere (Claflin et al., 2021; Vermeuel et al., 2023; Jensen et al., 2023).

The PTR-ToF-MS sampled the diluted GC eluent–zero air mixture at a rate of 120 cm$^3$ min$^{-1}$ through a polyether-ether-ketone (PEEK) capillary (25 mm, 0.25 mm ID) which directs the flow to the center of the focusing ion–molecule reactor (IMR). A separate flow of water-vapor-saturated air enters a pre-chamber where a plasma creates a reagent ion distribution that includes $H_3O^+$, water adducts (i.e., $H_3O(H_2O)_n^+$, where $n = 1, 2, 3$, etc.), and some amount of $O_2^+$ and $NO^+$ reagent ions that are considered impurities. These reagent ions from the pre-chamber enter the IMR alongside the eluent sample flow. There are two features of the Vocus PTR-ToF-MS discussed thus far that distinguish this instrument from other instruments that use $H_3O^+$ chemical ionization: (1) the Vocus PTR-ToF-MS uses a radio frequency (RF) only quadrupole around the IMR to generate $H_3O^+$ ions in excess by declustering water adducts of $H_3O^+$ and (2) the water vapor concentration in the IMR is estimated to be approximately 20 % by volume (Krechmer et al., 2018). We do not discuss the effects of IMR quadrupole voltage settings on PIDs here but instead point the reader to Li et al. (2024) for more information. We do not expect the differences in IMR quadrupole settings utilized in this study to explain the differences observed in the interlaboratory PID comparisons. The higher water vapor concentrations in the Vocus IMR are likely to have impacts that are unique to the Vocus PTR-ToF-MS for PIDs from VOCs historically affected by a water vapor dependence (e.g., formaldehyde, hydrogen cyanide, and formic acid) compared to PTR-MS instruments using a drift tube where water vapor concentrations are lower.

### 2.2.2 PID quantification from GC-PTR-ToF-MS measurements

For our method of quantifying PIDs, we use chromatographic separation prior to detection of product ions with PTR-ToF-MS. The advantage of using a GC when quantify-

**Table 2.** Lab-defined instrument settings for datasets contributed by each lab. Some labs provided data where the instrument was operated under different settings, and/or data were collected years apart, and thus we differentiate datasets by the letters a, b, and c.

| ID | IMR $T$ (°C) | IMR $P$ (mbar)[3] | $\Delta V_{IMR}$ (V) | $E/N$ (Td) | BSQ RF voltage (V) | $\Delta V_1$ (V) | $\Delta V_2$ (V) | Water flow (scm$^3$ min$^{-1}$)[4] | Inlet flow (cm$^3$ min$^{-1}$) | Date acquired |
|---|---|---|---|---|---|---|---|---|---|---|
| Lab 1a | 60 | 2.0 | 580 | 133 | 350 | −22.5 | −4.1 | 20 | 120 | May 2023 |
| Lab 1b | 60 | 2.0 | 580 | 133 | 300 | −22.5 | −4.1 | 20 | 120 | May 2024 |
| Lab 2a[1] | 60 | 2.4 | 575 | 110 | 300 | −29.0 | −7.3 | 19 | 100 | October 2020 |
| Lab 2b[1,2] | 60 | 2.4 | 660 | 126 | 400 | −4.4 | −8.1 | 20 | 100 | November 2023 |
| Lab 3a[2] | 100 | 1.5 | 365 | 125 | 215 | −39.7 | −4.5 | 20 | 96 | December 2020 |
| Lab 3b[2] | 100 | 1.5 | 385 | 133 | 215 | −32.0 | −4.0 | 15 | 88 | November 2022 |
| Lab 4 | 100 | 2.5 | 450 | 122 | 320 | −40.5 | −5.1 | 20 | 79 | September 2024 |
| Lab 5[2] | 110 | 2.5 | 624 | 131 | 250 | −27.5 | −3.5 | 21 | 180 | July 2021 |
| Lab 6a[2] | 90 | 1.5 | 480 | 160 | 255 | −19.1 | −6.5 | 15 | 260 | March 2021 |
| Lab 6b[2] | 90 | 1.5 | 480 | 160 | 255 | −19.1 | −6.5 | 15 | 290 | May 2022 |
| Lab 7a | 100 | 2.2 | 570 | 133 | 325 | −39 | −4.2 | 20 | 100 | April 2022 |
| Lab 7b | 100 | 2.2 | 570 | 133 | 325 | −39 | −4.2 | 20 | 100 | September 2022 |
| Lab 7c | 100 | 2.2 | 570 | 133 | 325 | −39 | −4.2 | 15 | 100 | May 2023 |

[1] Lab 2a and Lab 2b data come from two different instruments.
[2] IMR quadrupole RF voltage was 400 V. The IMR quadrupole RF voltage was 450 V for other instruments.
[3] 1 mbar = 100 Pa.
[4] Standard cm$^3$ min$^{-1}$ (standard conditions = 293.15 K and 101.325 kPa).

ing PIDs is that analytes in multi-component mixtures (e.g., calibration standards or ambient samples) can be separated before detection and thus avoid interference with PID quantification.

Figure 1 shows an example, using pentanoic acid, of the chromatographic method of determining PIDs from GC-PTR-ToF-MS measurements. As shown in Fig. 1a, we use a selected-ion chromatogram from the expected $H^+$ adduct ion signal to determine where to define the background and maximum signal mass spectra. The background mass spectrum is subtracted from the signal mass spectrum to create the isolated mass spectrum shown in Fig. 1b. The high-resolution fitted peak areas of each product ion $m/q$, with at least 1 % contribution to the isolated mass spectrum, are added together to represent the sum product ion signal, and the relative contribution of each ion to the sum represents the PID. As shown in Fig. 1b, some analytes had ions that made small contributions ($< 5$ %) to the isolated mass spectrum in addition to the ions that were included in the PID for pentanoic acid. If ions could not reasonably be explained mechanistically as product ions from the target analyte and made small contributions ($< 5$ %) to the isolated mass spectrum, we omitted them in the determination of a PID.

## 2.3 PID measurement as a function of instrument settings

In the PTR-ToF-MS instruments in this study, chemistry that forms PIDs occurs in the IMR immediately downstream of the capillary that serves as the sample inlet for the instrument (Fig. 2). In the IMR a voltage differential ($\Delta V_{IMR}$ in Fig. 2) creates an electric field that focuses ions through the reactor. However, the electric field ($E$, V m$^{-1}$) strength the ions experience is reduced by the reactor air number density ($N$, molec. cm$^{-3}$). The influence of the reduced electric field strength, $E/N$, on $H_3O^+$ ion chemistry is well documented in PTR-MS literature for both drift tube (Yuan et al., 2017) and ion–molecule reactors (Krechmer et al., 2018) and can be calculated following Eq. (1) (De Gouw and Warneke, 2007):

$$\frac{E}{N} = \frac{\Delta V_{IMR} \cdot T \cdot R}{L_{IMR} \cdot P \cdot Av \cdot 10^{-21}} , \qquad (1)$$

where $\Delta V_{IMR}$ is the voltage differential between the IMR back and front (V), $T$ is the IMR temperature (K), $R$ is the ideal gas constant ($8.3 \times 10^{-2}$ m$^3$ kPa K$^{-1}$ mol$^{-1}$), $L_{IMR}$ is the length of the IMR (10 cm for the instruments in this study), $P$ is the IMR pressure (kPa), $Av$ is Avogadro's number, and $10^{-21}$ is a conversion factor from V m$^{-2}$ to units of townsend (Td). We note that for the Vocus instruments discussed here the RF-only quadrupole around the IMR adds to the electric field strength, an effect that is not accounted for in this equation. Li et al. (2024) showed that although the IMR RF voltage can affect analyte sensitivity, it did not affect PIDs. All the instruments in this study operated with similar RF voltages for the IMR (between 400 and 450 V), so we exclude this contribution from the $E/N$ values we report. To measure the effects of $E/N$ on select PIDs in this study, we varied the pressure in the IMR – while keeping the reactor voltage differential ($\Delta V_{IMR}$) constant – between 1.4 mbar (0.14 kPa) and 3.0 mbar (0.30 kPa), resulting in $E/N$ values ranging from 90 to 190 Td.

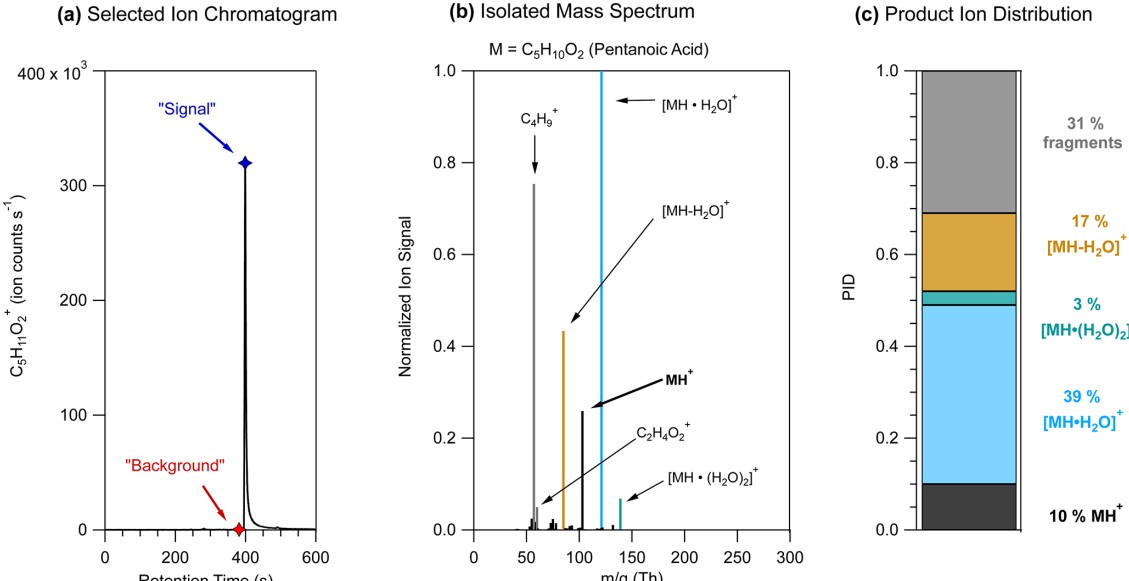

**Figure 1.** Steps of a method for determining PIDs using pentanoic acid as an example. **(a)** The selected-ion chromatogram for the expected $H^+$ adduct of pentanoic acid, $C_5H_{11}O_2^+$, showing ion signal as a function of retention time. Markers show the retention time when the maximum signal (blue) and background (red) mass spectra were defined. **(b)** The pentanoic acid isolated mass spectrum is determined by subtracting the background mass spectrum from the maximum signal mass spectrum. Ion signals are normalized to the highest ion signal. **(c)** Product ion distribution (PID) measured from the isolated mass spectrum for pentanoic acid using data from panel **(b)**.

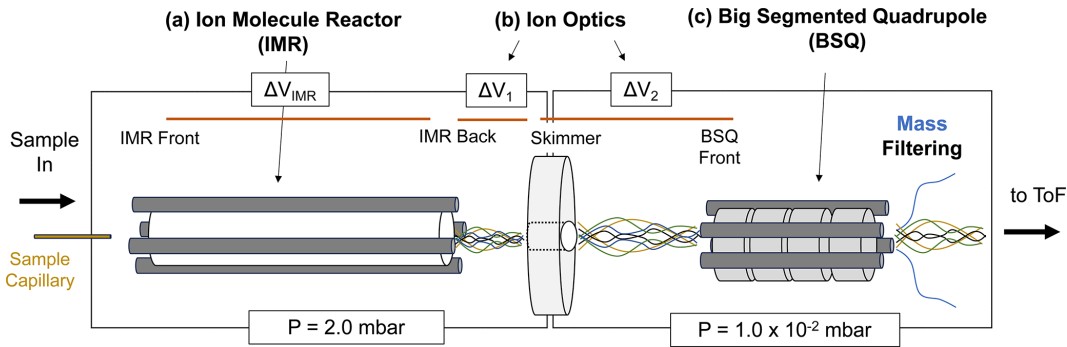

**Figure 2.** Simplified diagram of the front end of the PTR-ToF-MS evaluated in this study. Sample air enters the instrument through a capillary and is directed to the IMR. **(a)** The IMR voltage difference between the back and front ($\Delta V_{IMR}$) in part controls the energy of ion collisions. **(b)** After the IMR, there are two sections of the ion trajectory with voltage differentials that occur at relatively high pressures; these are between the transfer optics (skimmer–IMR back, $\Delta V_1$; BSQ front–skimmer, $\Delta V_2$) as shown. **(c)** The big segmented quadrupole (BSQ) is an RF-only quadrupole that filters ions acting as a high-pass filter. Pressures for the regions defined by the boxed areas are shown at the bottom of the figure (1 mbar = 100 Pa).

Although PIDs are initially formed in the IMR, $m/q$-dependent transmission efficiencies between the IMR and the time-of-flight mass analyzer can affect the PIDs that are ultimately measured (Jensen et al., 2023; Li et al., 2024). We isolate three parts of the ion trajectory in the instrument as possible locations for affecting PIDs through collisional dissociation, quadrupole mass filtering, and/or other transmission effects. The first two areas where ions may undergo declustering of water adducts or collisionally induced fragmentation are shown in Fig. 2 as $\Delta V_1$ and $\Delta V_2$, which correspond to

the voltage differential between the skimmer and IMR back ($\Delta V_1$) and the BSQ front and skimmer ($\Delta V_2$). These ion optic voltage differences have been demonstrated to contribute to declustering reactions in a similar mass spectrometer (Brophy and Farmer, 2016).

In this study, we vary the voltage difference between each ion optic component relationship following the methodology of previous studies (Brophy and Farmer, 2016; Lopez-Hilfiker et al., 2016) by incrementally changing the entire set of voltages upstream (i.e., in the direction of the inlet) of

the tested component relationship. We performed these ensemble voltage changes manually without the use of tuning software. The range of tested voltages is based on the observed voltage differences in the interlaboratory comparison dataset. For $\Delta V_1$ we measured PIDs as a function of $\Delta V$ ranging from $-3$ to $-50$ V, and for $\Delta V_2$ we tested a range of $-1$ to $-10$ V. We performed these PID sensitivity tests on the instrument corresponding to Lab 1. The skimmer component in the $\Delta V_1$ and $\Delta V_2$ relationships described here corresponds to the skimmer located right before the BSQ (i.e., not the skimmer 2 component also present in all versions of the Vocus instrument evaluated here).

The third ion optic component we evaluate is the effect of the RF amplitude voltage of the big segmented quadrupole (BSQ) in filtering ions of different $m/q$. The primary function of the BSQ is to act as a high-pass filter limiting the transmission of lower-mass reagent ions (i.e., $H_3O^+$ $m/q = 19.02$ Th and $(H_2O)H_3O^+$ $m/q = 37.03$ Th) to the detector and thus extending the lifetime of the detector (Krechmer et al., 2018). Product ions with an $m/q$ in the range of these major reagent ions will also experience decreased transmission (Jensen et al., 2023; Li et al., 2024). We measured PIDs at nine different BSQ voltage settings between 225 and 450 V. Although we focus on three areas where ion $m/q$-dependent transmission effects may occur, we note that mass discrimination effects can occur elsewhere in the instrument and for other reasons such as detector degradation (Heinritzi et al., 2016) or discrimination of higher $m/q$ ions because of other quadrupole transmission effects (Holzinger et al., 2019; Antony Joseph et al., 2018).

### 2.4 PID measurement as a function of sample capillary insertion distance

A small PEEK (25 mm length, 0.18 mm inner diameter) capillary, secured by two Viton O-rings, serves as the sample inlet to the instrument. The distance that this capillary is inserted into the instrument can be manually changed and impacts the ionization chemistry that occurs immediately at the exhausting end of the capillary. We characterized the effects of the capillary insertion distance on the measured PID from pentanoic acid by turning off all voltages to the IMR, closing the standby valve between the IMR region and the rest of the instrument, and manually adjusting the capillary to a different insertion distance. With the capillary at the desired insertion distance, we returned the IMR to previous operating conditions and acquired a GC measurement of pentanoic acid. We then changed the capillary insertion distance between 3 and 13 mm for five total measurements.

### 2.5 Interlaboratory comparison of PIDs

We compare PIDs from seven different instruments under lab-defined settings. Lab-defined settings for all instruments are shown in Table 2.

### 2.6 Restroom air measurement

To demonstrate the uncertainties introduced by interfering product ions in ambient air, we deployed our GC-PTR-ToF-MS to a restroom as detailed in Link et al. (2024). Briefly, the restroom air sample was acquired during a weekend-long measurement period. The restroom air contained elevated concentrations of terpenoids (i.e., monoterpenes, monoterpene alcohols, and monoterpene acetate esters) that reacted with ozone and created oxygenated VOC products. The relative VOC composition of the restroom air stayed consistent over the measurement period with concentrations decreasing from the start of the period to the end. We highlight one GC chromatogram acquired during that measurement period to demonstrate the effect of PIDs on ion attribution from an indoor air sample.

### 2.7 Data processing

During GC measurements mass spectra were collected at a rate of 5 Hz. Mass calibration, resolution and average peak shape determination, and high-resolution peak fitting were all performed in Tofware v3.2.5 (Aerodyne Research Inc.). Mass accuracy was maintained within $\pm 6$ ppm when performing mass calibrations. A peak list containing 1046 ions was used for high-resolution peak fitting. VOCs present in calibration standards were used to inform what product ions were likely to be expected following the definitions in Table 1. Selected-ion chromatograms and isolated mass spectra were produced using the analysis tools in TERN v2.2.20 software (Aerodyne Research Inc.). Ion signals were not ToF duty cycle corrected.

## 3 Results and discussion

### 3.1 Influence of instrument configuration on PIDs

#### 3.1.1 Influence of IMR $E/N$ on PIDs

IMR $E/N$ is an important determinant of water clustering and fragmentation. Figure 3 shows the PID for pentanoic acid, ethanol, and toluene measured at different $E/N$ values.

We highlight pentanoic acid because it forms fragments and water clusters across a wide $m/q$ range ($m/q$ 39.02 to $m/q$ 139.10). We highlight ethanol because it forms water clusters and a hydride transfer product. We highlight toluene because it forms charge and hydride transfer products as well as a product we classify as "other" ($C_6H_7O^+$). In the case of pentanoic acid, the contribution of the $H^+$ adduct increased from 0.26 to 0.47 with increasing $E/N$ (Fig. 3). This change in the $H^+$ adduct contribution was mostly due to the decreasing contribution of the first water cluster from 0.53 at the lowest $E/N$ to 0.06 at the highest $E/N$. In contrast, the contribution of total fragmentation products (dehydration + other fragment ions) increased from 0.20 at the lowest $E/N$ to 0.60

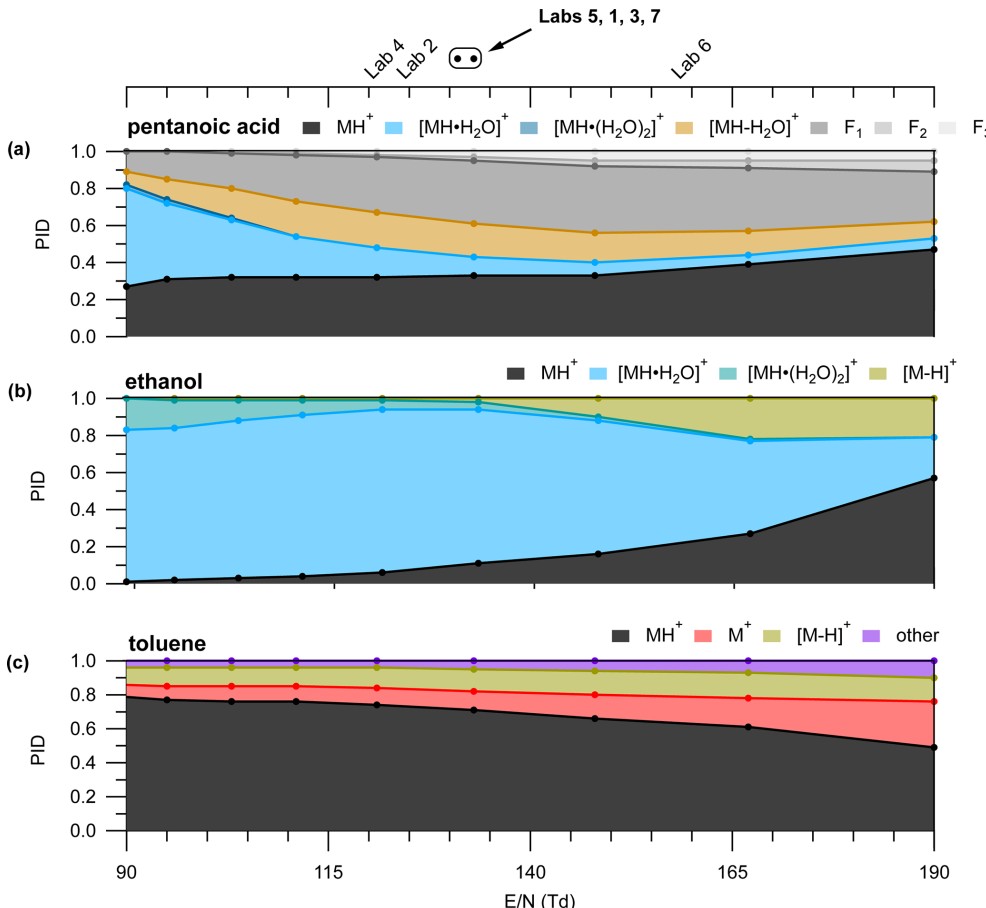

**Figure 3. (a)** Pentanoic acid PID as a function of $E/N$. Color labels in the legend above the panel correspond to the colored traces in the panel. PIDs for **(b)** ethanol and **(c)** toluene as a function of $E/N$. $E/N$ values used by the different labs in the interlaboratory comparison are shown in the top axis. The circle markers indicate values where the lab text markers would overlap and are listed in order of $E/N$ in the corresponding text label. Measurements were acquired with a BSQ voltage of 300 V.

at an $E/N$ of 148 Td (Fig. 3). Above $E/N$ 148 Td, the contribution of the $H^+$ adduct to the PID increases and the relative contribution of fragment ions decreases. The general pattern of the water cluster and fragment product ion variation with $E/N$ shown in Fig. 3 suggests a lower $E/N$ will decrease the contributions of fragment ions in the mass spectrum. However, higher $E/N$ values will decrease the contribution of water clusters to the mass spectrum. Because different PIDs (i.e., different contributions of fragments, water clusters, and the $H^+$ adduct) are generated at the different values of $E/N$ tested here, measurable product ion formation will likely occur for a variety of VOCs regardless of $E/N$. As is the case for the three VOCs highlighted here, secondary product ion generation is not suppressed across the tested $E/N$ range.

As another example, we show (Fig. 3b and c) how the PIDs vary as a function of $E/N$ for species that can generate product ions from reactions with impurity reagents $NO^+$ and $O_2^+$. Impurity reagent ions are generated unintentionally in the PTR-ToF-MS and result from oxygen ionizing in the ion source plasma. We show here, using ethanol and

toluene as examples, that a higher $E/N$ may qualitatively indicate that a user could expect more important contributions of hydride and charge transfer products to the PID. Ethanol forms $C_2H_5O^+$, a likely hydride transfer product from reaction with $NO^+$, while toluene forms $C_7H_7^+$, a likely hydride transfer product from reaction with $NO^+$ (Smith et al., 2020), and $C_7H_8^+$, a charge transfer product from reaction with both $O_2^+$ and $NO^+$ (Coggon et al., 2024; Koss et al., 2016). The increased contributions of charge and hydride transfer products to the PIDs of ethanol and toluene potentially suggest an increased influence of impurity reagent ions, but we do not have an explanation for how impurity reagent ion concentrations would increase with increasing $E/N$ in the IMR. We note that the presence of air leaks in the reagent delivery system may increase the importance of impurity reagent ion chemistry. Also, purging the water reagent source with pure nitrogen may be a possible method to decrease impurity reagent ion chemistry due to the presence of dissolved oxygen.

### 3.1.2 Influence of BSQ RF voltage on PIDs

Another important influence on PIDs is the BSQ RF amplitude voltage (referred to hereafter as "BSQ voltage"). BSQ voltages observed from the lab-defined settings in the interlaboratory comparison dataset ranged from 215 to 400 V. The BSQ acts as a high-pass filter, and thus low-mass ion transmission decreases with increasing BSQ voltage. In other words, at low BSQ voltages (e.g., 225 V) we would expect to see greater transmission of low-mass ions (e.g., $m/q < 55.04$ Th) compared to higher voltages (e.g., 450 V). When considering how the BSQ affects PIDs, we expected that product ions that had low mass, both the $H^+$ adduct and fragment ions, would be most affected by different BSQ voltages versus the higher $m/q$ water cluster products.

Figure 4 shows the ion signals and PIDs for pentanoic acid measured across a range of BSQ voltages at an $E/N$ of 133 Td. The integrated ion counts in Fig. 4 demonstrate the effect of the BSQ voltage on total transmission of ions, whereas the PIDs demonstrate transmission effects relative to other ions. Because the BSQ mainly acts as a high-pass filter, BSQ effects on PIDs are likely to be most pronounced for VOCs that generate lower $m/q$ ions like the fragment ions generated from pentanoic acid. The contributions of fragment ions to the PID for pentanoic acid are most pronounced at BSQ voltages less than 350 V. As the BSQ voltage increases, the lowest $m/q$ product ion ($C_3H_5^+$, $C_3H_5^+$, $C_4H_9^+$, and $C_2H_4O_2^+$) contributions decrease. At 450 V the $C_3H_3^+$ and $C_3H_5^+$ ions no longer make measurable contributions to the PID, and the contribution of $C_4H_9^+$ has decreased by a factor of 5. However, as the contribution of lower $m/q$ ions to the PID decreases with increasing BSQ voltage, the contribution of higher $m/q$ ions ($H^+$ adduct and water clusters) generally increases for pentanoic acid. The relative contribution of the single water cluster to the PID increases by a factor of 6 at 450 V compared to 225 V. Notably, we cannot explain why the integrated ion counts for the $MH^+$ ion from pentanoic acid decrease going from a BSQ voltage of 200 to 300 V.

### 3.1.3 Influence of ion optic voltages and capillary distance on PIDs

We found that ion optic voltage differences (i.e., $\Delta V_1$ and $\Delta V_2$ in Fig. 2) and the capillary insertion distance did not impact the pentanoic acid PID as clearly as $E/N$ and the BSQ settings. Figures presented in the Supplement demonstrate the variability in PIDs measured for pentanoic acid when testing the voltage differences for $\Delta V_1$ (Fig. S2 in the Supplement) and $\Delta V_2$ (Fig. S3), as well as the sample capillary insertion distance (Fig. S5). We also analyzed the PID for benzene to investigate if charge transfer product ions were modulated by the capillary distance. We did not observe any clear trends in the PID for pentanoic acid or the charge trans-

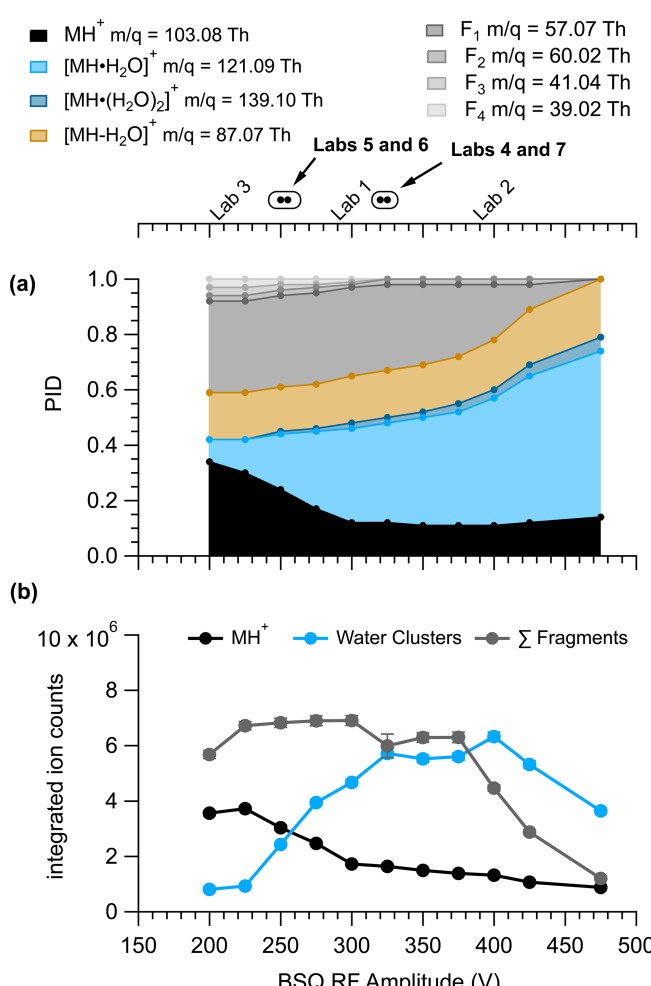

**Figure 4.** Pentanoic acid **(a)** PID and **(b)** product ion signals as a function of BSQ RF amplitude voltage measured with IMR $E/N = 133$ Td. Because the BSQ is supposed to mainly act as a high-pass filter, the $m/q$ values for the product ions are listed next to the product ion definition in the legend to contextualize $m/q$-dependent transmission effects from the changing BSQ voltage. The ion signals for the $MH^+$ ion, sum of the water cluster product ions, and sum of the fragment product ions were determined by integrating product ion peaks from their selected-ion chromatograms. Error bars are difficult to visualize but show the error from the residual peak area. The BSQ voltages used by the laboratories in the comparison are shown in the top axis. The circle markers indicate values where the lab text markers would overlap and are listed in order of BSQ voltage in the corresponding text label.

fer product ion contributions to the benzene PID as a function of capillary distance.

Although we did not observe major effects of $\Delta V_1$ and $\Delta V_2$ on the pentanoic acid PID, we did observe notable changes in the PIDs for other VOCs as shown in Fig. 5.

Changes in PIDs induced by voltage gradients across the ion optics likely result from collisionally assisted fragmentation and declustering. As shown in Fig. 5 we observe in-

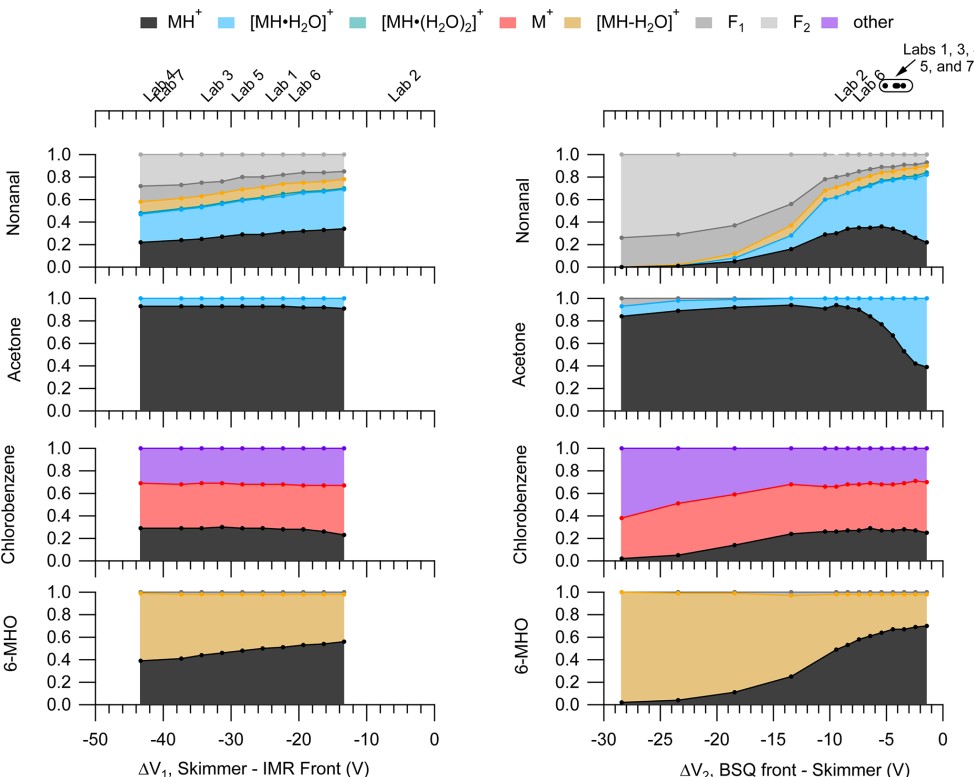

**Figure 5.** PIDs for nonanal, acetone, chlorobenzene, and 6-methyl-5-hepten-2-one (6-MHO) as a function of $\Delta V_1$ (left) and $\Delta V_2$ (right). The top axes for the left and right panels correspond to the bottom axes, and the midpoint of the labels show the $\Delta V$ corresponding to the respective lab. Circle markers on the top-right axis correspond to a range of $\Delta V$ of $\pm 1$ V and the text labels shown above for clarity. These PIDs were measured at an IMR $E/N$ of 150 Td and a BSQ voltage of 300 V. Figure S4 shows these PIDs measured at an IMR $E/N$ of 106 Td.

creased fragmentation and increased water adduct declustering as the absolute $\Delta V$ increases for both $\Delta V_1$ and $\Delta V_2$. These changes in the PIDs are associated with the increased energy of ion collisions as they traverse the voltage gradient. These collisional effects are highlighted in the PIDs for nonanal and 6-MHO where fragmentation product ion contributions to the PIDs increase with increasing $\Delta V$.

The PID for chlorobenzene consists of the $H^+$ adduct, a charge transfer product, and another product ion formed by an unknown mechanism, $C_6H_7O^+$. Compared to nonanal and 6-MHO the PID for chlorobenzene does not show as strong of an influence of ion collisions changing the PID. The relative stability of the chlorobenzene PID with $\Delta V$ for both $\Delta V_1$ and $\Delta V_2$ suggests that other species that have PIDs mostly containing charge transfer and hydride transfer product ions may also be minimally influenced by ion optic voltage differences. However, the increasing contributions of both $C_6H_7O^+$ (the "other" product ion) and $C_6H_5Cl^+$ (the charge transfer product ion) to the chlorobenzene PID with increasing $\Delta V_2$ possibly suggest collisions may be important for converting the $H^+$ adduct to these other product ions given high enough collisional energy.

We did not observe major effects of ion optic voltage differences on the pentanoic acid PID, but the results in Fig. 5 suggest that increased ion optic voltage differences may increase the contribution of fragmentation and decrease the contribution of water cluster ions to a PID for other molecules. The voltage differences used by the different labs included in the interlaboratory comparison encompassed a smaller range for $\Delta V_2$ compared to $\Delta V_1$.

We observe sensitive changes to the nonanal and 6-MHO PIDs within the narrow range of voltages used for $\Delta V_2$ but also measurable, albeit less sensitive, changes in the PIDs for $\Delta V_1$. Although the effects of $\Delta V_1$ on PIDs were not as sensitive as $\Delta V_2$, we acknowledge the potentially important role this ion optic voltage difference could have in interpreting differences in PIDs measured between labs such as Labs 4 and 6, in the interlaboratory comparison, which have a difference in $\Delta V_1$ between the two labs of approximately 20 V. For instance, going from the highest measured $\Delta V_1$ we measured for 6-MHO to the lowest $\Delta V_1$, the contribution of the $MH^+$ product ion to the PID decreases by 30 % (i.e., from 0.59 to 0.36). Because of the greater sensitivity of the PIDs to $\Delta V_2$, we highlight the importance of this relationship in affecting PIDs but note that Fig. 5 demonstrates that differ-

ences in $\Delta V_1$ are likely important enough to create differences in product ion contributions to PIDs on the order of 10 % to 30 % for the instruments evaluated as part of the interlaboratory comparison.

An important implication of sensitive declustering and fragmentation effects from $\Delta V_2$ is that the IMR $E/N$ alone cannot accurately predict the extent of possible fragmentation or declustering affecting PIDs. We show in Fig. 6 how the PID for acetone and nonanal changes when varying the IMR $E/N$, $\Delta V_2$, and BSQ voltage individually compared to a reference set of instrument operating parameters (dotted red line corresponding to $E/N = 135$ Td, $\Delta V_2 = -8.5$ V, and BSQ RF voltage = 300 V). For both acetone and nonanal, we see the same effects of increasing water cluster declustering and fragment ion formation as $E/N$ goes from low to high values (Fig. 6a and d) as we observed for pentanoic acid (Fig. 3). While keeping the IMR $E/N = 135$ Td and varying $\Delta V_2$, we see changes in the nonanal PID (Fig. 6e) that are nearly as pronounced as similar incremental changes in the IMR $E/N$. For instance, at a $\Delta V_2 = -4.4$ V the PID for nonanal is similar to the PID measured at 100 Td. To a rough approximation, a 1 V change in $\Delta V_2$ is equivalent to a change in IMR $E/N$ of 9 Td for nonanal. A similar sensitivity to $\Delta V_2$ is observed for acetone, but our interpretation is limited because the PID only has a minor contribution from the water cluster under all conditions. In contrast to pentanoic acid (Fig. 4), major PID changes for acetone and nonanal were not observed when scanning the BSQ RF voltage, demonstrating that the combined influence of the instrument components evaluated here on measured PIDs can vary considerably between different chemical species.

## 3.2 Interlaboratory comparison of PIDs

We compare PIDs measured from the seven laboratories under lab-defined settings. Acetonitrile and $\alpha$-pinene were the only VOCs with PIDs measured by every lab. We highlight select VOCs with a particular propensity for water cluster and/or fragment ion formation that were commonly measured amongst the labs for a qualitative comparison. We then compare a more diverse suite of VOCs for a quantitative characterization of PIDs across instruments.

### 3.2.1 Qualitative comparison of PIDs across instruments

Figure 7 highlights differences in PIDs measured from select VOCs common across most of the instruments. The appearance and contribution of product ions to the PID of a given VOC varied between instruments but can mostly be qualitatively explained by variations in $E/N$, $\Delta V_2$, and BSQ voltage. We note that the effects of instrument configuration (i.e., $E/N$, BSQ voltage, ion optic voltages) should have predictable effects on PIDs measured by a single instrument, and thus using the product ion quantification methods described

later in Sect. 3.5 is not dependent on our ability to reconcile instrument-to-instrument differences.

Data shown in Fig. 7 originate from instruments operating within a relatively narrow range of $E/N$ (122 to 133 Td), with the exception of Lab 6, which ran at an $E/N$ of 160 Td, and the ethyl acetate measurement from Lab 2. Our analyses of pentanoic acid PID variability as a function of instrument configuration provide some context for interpreting the PID variability observed here. Measurements of the pentanoic acid PID as a function of $E/N$ in Fig. 3 demonstrate that variability in water cluster and fragment product ion contributions to the PID may vary on the order of approximately 10 % when comparing measurements acquired at an $E/N$ of 120 Td versus 130 Td. Similarly, we may expect variability of water cluster contributions for the VOCs shown in Fig. 7 to vary on the order of 10 % within the $E/N$ range of all labs except Lab 6. Water clusters made some contribution to the PID for at least one of the VOCs for each lab except Lab 6, which operated at the highest $E/N$ (160 Td).

We expected the acetone PID could provide evidence of BSQ low-mass filtering as the $m/q$ of the $H^+$ adduct ion ($m/q$ 59.05 Th) is lower than the water cluster product ion ($m/q$ 77.06 Th), and so lower BSQ voltages may correspond to higher contributions of the $H^+$ ion to the PID compared to the water cluster. Comparison of the acetone PID from Lab 1 versus Lab 2 and Lab 7 displays the opposite trend where, when BSQ voltage increases, the contribution of the $H^+$ ion increases compared to the water cluster ion. For Lab 2, we suspect this discrepancy in BSQ effect is explained by the mechanism of acetone water clusters formed in the IMR likely declustering after passing through the $\Delta V_2$ ion optic relationship (with the highest $\Delta V_2 = -8.1$ V indicating potentially important fragmentation/declustering), creating a measured PID entirely consisting of the $H^+$ adduct. However, we do not have an explanation for why Lab 7 does not show water cluster contributions to the acetone PID, where Lab 1 shows about a 10 % contribution, despite having nearly identical settings to the Lab 1 instrument. This comparison of the acetone PID with BSQ voltage demonstrates the challenge of generalizing patterns of PIDs from a single instrument setting to other instruments.

Each instrument in this intercomparison was operated with a different BSQ voltage which likely influenced variability in PIDs between instruments. For several of the VOCs in Fig. 7 we might expect higher contributions of water clusters to the PIDs for acetonitrile, ethanol, and acetone at higher BSQ voltages because higher voltages decrease the transmission efficiency, relative to water clusters, for the $H^+$ adduct. For instance, Lab 3 operated with a BSQ voltage of 215 V, and Lab 2 operated with a voltage of 400 V, representing the lower and upper ends, respectively, of the dataset BSQ voltage range. One possible explanation for the difference in the water cluster contribution to the acetonitrile PID of 3 % and 24 % measured for Lab 3 and Lab 2, respectively, is increased relative transmission efficiency of the water cluster at

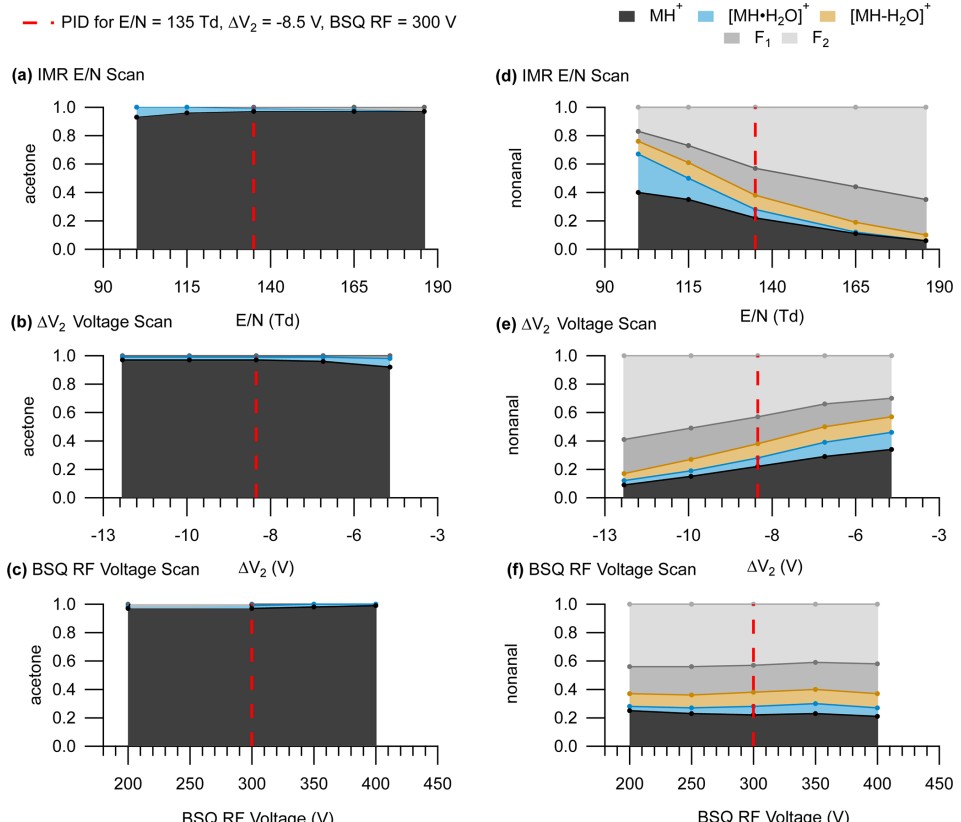

**Figure 6.** PIDs for acetone (left panels) and nonanal (right panels). Panels **(a)** and **(d)** show PIDs as a function of IMR $E/N$, panels **(b)** and **(e)** show PIDs as function of $\Delta V_2$, and panels **(c)** and **(f)** show PIDs as a function of BSQ RF voltage. The dotted red line shows where the settings for the IMR, $\Delta V_2$, and the BSQ were equivalent ($E/N = 135$ Td, $\Delta V_2 = -8.5$ V, and BSQ RF = 300 V). Because PIDs are more sensitive to $\Delta V_2$ compared to $\Delta V_1$, we only show the PIDs as a function of $\Delta V_2$ here for simplicity.

the higher BSQ voltage used in Lab 2 (both labs have similar $E/N$).

Ethyl acetate was also impacted by BSQ voltage effects (Fig. 7). The $E/N$ for the Lab 3 ($E/N = 122$ Td) mea-
5 surement of ethyl acetate falls in between that of Lab 1 ($E/N = 133$ Td) and Lab 2 ($E/N = 110$ Td), and thus we might expect the PID to be similar to those two labs. In contrast to Labs 1 and 2, the Lab 3 ethyl acetate PID shows a higher contribution of fragment ions and does not show a
10 water cluster contribution. The two major fragment ions for ethyl acetate ($C_2H_3O^+ = 43.02$ Th and $C_2H_5O_2^+ = 61.03$ Th) are similar in $m/q$ to the fragment ions of pentanoic acid ($C_3H_5^+ = 41.04$ Th and $C_4H_9^+ = 57.07$ Th) that we saw affected by the BSQ voltage in Fig. 4. Thus, the lower BSQ
15 voltage used for Lab 3 (BSQ = 215 V), compared to Labs 1 (BSQ = 300 V) and 2 (BSQ = 400 V), likely increased the transmission efficiency of fragment ions, relative to the $H^+$ adduct and water cluster, and increased their contribution to the PID for Lab 3.
20 Of the VOCs presented here, $\alpha$-pinene, shows considerable fragmentation but also reasonable agreement in the PID ($\pm10$ % for any given product ion contribution to the

PID) across instruments. Variability in $\alpha$-pinene PIDs between instruments can be qualitatively explained by differences in $E/N$. Lab 6, operating at an $E/N$ of 160 Td (higher 25 fragmentation than the other instruments), showed a near-equal contribution of the $H^+$ adduct, $F_1$, and the sum of other fragments to the PID, whereas the other instruments showed roughly half the $H^+$ adduct and half of $F_1$, with some ($< 10$ %) contribution of the sum of other fragments. We ex- 30 pect $\alpha$-pinene and most other monoterpenes to be minimally influenced by changes in BSQ voltage (and thus low-mass-filtering effects) as most of the major product ions are greater than $m/q$ 55.04 Th (corresponding to the reagent ion double water cluster, $(H_2O)_2H_3O^+$) where mass-filtering effects are 35 expected to be less pronounced (Krechmer et al., 2018).

Reagent ion impurities, $O_2^+$ and $NO^+$, are likely responsible for charge and hydride transfer product ions observed for benzene and ethanol shown in Fig. 7. In Fig. 3 we show that the PID contribution for both hydride (as seen for ethanol and 40 toluene) and charge transfer products (as seen for toluene) increases with increasing $E/N$. However, variability in $E/N$ does not explain the differences in hydride transfer product contributions to the PID for ethanol and charge trans-

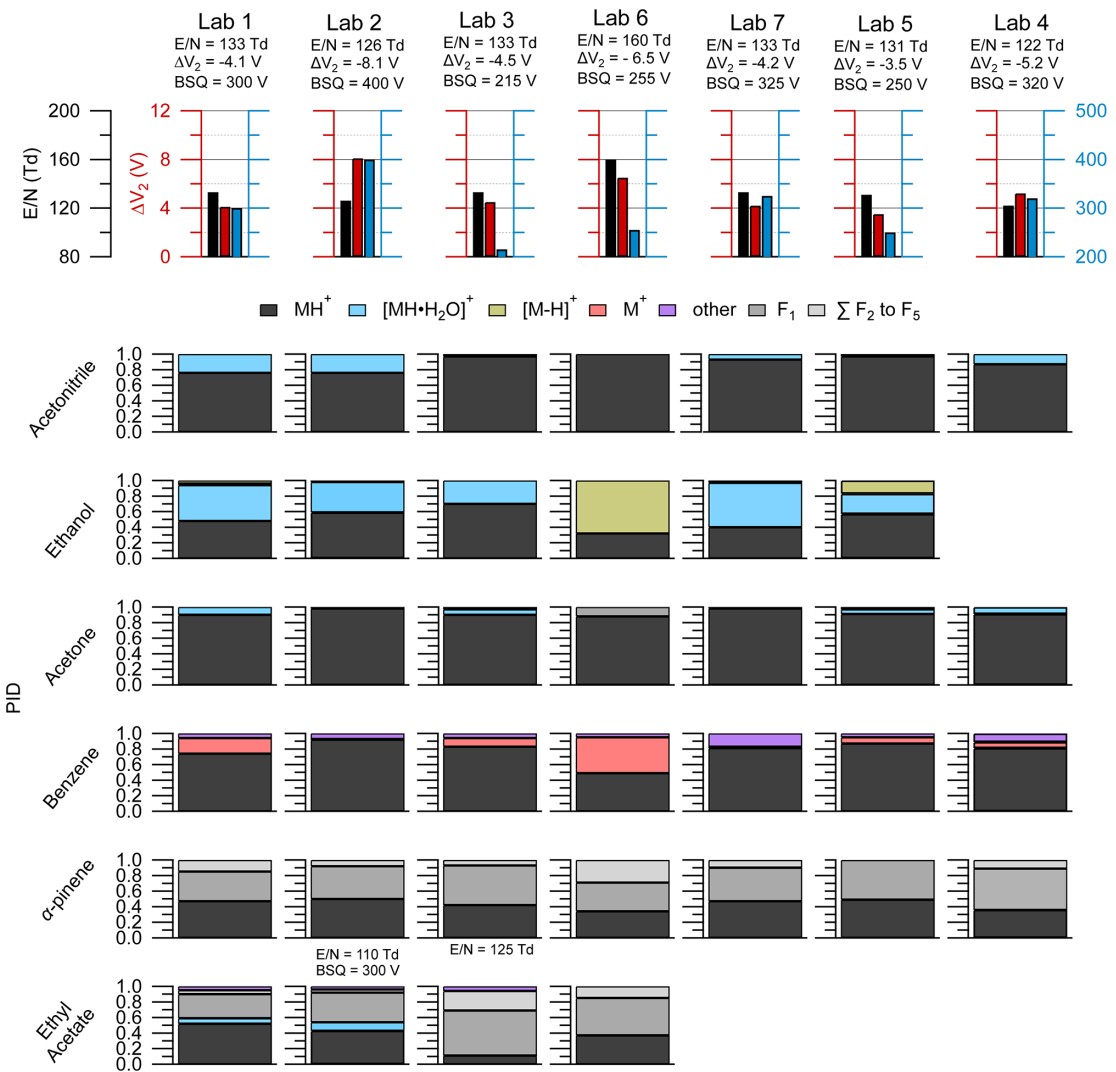

**Figure 7.** The top row shows the lab identity label (i.e., Lab 1, Lab 2, etc.) and corresponding $E/N$ (left axis, black), $\Delta V_2$ (left axis, red), and BSQ voltages (right axis; blue) used for the PID measurements shown below. PIDs are shown in the lower panels for select VOCs from the interlaboratory comparison dataset and were chosen based on if the VOC measurement was available for each lab. Empty spots where a bar plot would be indicate that lab did not have measurements for the VOC in the corresponding row. The PIDs for ethyl acetate were measured for Lab 2 and Lab 3 under slightly different instrumental conditions than the rest of the VOCs, and the corresponding $E/N$ and BSQ voltages are shown above the bar plots. Contributions of 3 % or less to the PID may be difficult to see in the figure, but exact values can be found in the H₃O⁺ PID library.

fer product contributions to the PID for benzene between the labs in Fig. 7. Lab 6, which operated with the highest $E/N$ (160 Td), had the largest contributions of both the hydride transfer product for ethanol and the charge transfer prod-
5 uct for benzene, which is consistent with the observation of more impurity reagent ion chemistry at a higher $E/N$. However, Lab 1 and Lab 7 operated with nearly the same $E/N$, $\Delta V_2$, and BSQ voltage, but Lab 7 did not measure the charge transfer product for benzene, whereas Lab 1 measured
10 a 20 % contribution. We hypothesize that increased inlet flow rates increase $O_2^+$ and/or $NO^+$ chemistry as evidenced by the ethanol hydride transfer product making the largest contribu-

tions to the ethanol PID for Lab 5 and Lab 6, which operated their instruments at higher flow rates compared to the other labs (Lab 5 = 180 cm³ min⁻¹ and Lab 6 = 290 cm³ min⁻¹,  15 while the other systems operated with an inlet flow rate of approximately 100 cm³ min⁻¹). The increased inlet flow rate may increase mixing of sample air and dilute the water-vapor-saturated air in the ionization region, thus generating more $NO^+$ and $O_2^+$ reagent ions.  20

We note that several aromatics (e.g., benzene, toluene, chlorobenzene) also generated a product ion, $C_6H_7O^+$, that we could not identify a mechanism for and we classified as "other". With regard to benzene detection, this product ion

contributed 20 % to the PID for Lab 7 but made smaller contributions ($< 5\%$) to the PIDs for other labs. In the case of Lab 7, larger contributions of $C_6H_7O^+$ did not coincide with enhanced contributions of the charge transfer product to the benzene PID, so this ion may not be a product of $O_2^+$ and/or $NO^+$ chemistry. Because $C_6H_7O^+$ is generated from several aromatics (see $H_3O^+$ PID library), it may be an important isobaric interference for phenol.

### 3.2.2 Quantitative comparison of PIDs across instruments

We calculated the average and standard deviation of the mean of the product ion contributions to the PIDs for 12 VOCs contained within the interlaboratory comparison dataset (Fig. 8). In contrast to the reporting uncertainties later discussed in Sect. 3.6, these averages and standard deviations are meant to quantitatively show variability across the instruments in this study. Many of the VOCs had standard deviations ($1\sigma$) for product ion contributions to PIDs that varied by no more than 0.30, thus providing a constraint for predicting PIDs across instruments operating under different conditions. Generally, the relative standard deviation (RSD) of product ion contributions to PIDs was larger for product ions with smaller fractional contributions (e.g., $< 0.10$) compared to larger contributions (e.g., $> 0.30$). For instance, the average and standard deviation of the contribution of the $MH^+$ ion to the methyl ethyl ketone PID was $0.90 \pm 0.06$ (7 % RSD), whereas the water cluster was $0.08 \pm 0.06$ (75 % RSD). Ethanol and acetonitrile showed considerable (i.e., $> 40\%$ RSD) product ion variability (Fig. 8). For ethanol, the importance of the water cluster was highly dependent on $E/N$. Additionally, the fraction of the hydride transfer product ranged from $< 0.05$ to roughly 0.50. The ethanol and acetonitrile PIDs are not only influenced by $E/N$ but also likely impacted by the BSQ voltage since the $H^+$ adducts have a relatively low $m/q$ (i.e., $m/q < 55.04$ Th). VOCs like isoprene and the aromatics have PIDs that are impacted by $NO^+$ and $O_2^+$ reagent ion chemistry, which, as discussed above, is difficult to predict without directly measuring PIDs of susceptible VOCs. The general trend of fragmentation/declustering with increasing $E/N$ and $\Delta V_2$ can be used as a guideline to inform a user about how they might expect their PIDs to deviate from the averages shown in Fig. 8. We recommend the $H_3O^+$ PID library as a guide for estimating PIDs for VOCs measured with Vocus PTR-ToF-MS instruments in the absence of direct measurements.

### 3.2.3 Consistency of PIDs measured over time

Two labs, Lab 6 and Lab 7, provided data where the instrument was operated under the same voltage configurations, but PIDs were measured a year or more apart. Figure 9 shows the variability in PIDs for four select VOCs from these two labs over a year.

Measurements from both labs indicate that, given the same voltage configurations on the same instrument, PIDs can change over time. The largest change from the subset of VOCs in Fig. 9 is the water cluster contribution to the ethanal (acetaldehyde) PID, from Lab 7, starting at 24 % and decreasing to 4 % after 13 months. Isoprene from Lab 7 has fragment and charge and/or hydride transfer product ions that appear in the PID after 5 months.

The PIDs for the four VOCs from Lab 6 show greater contributions of fragment and charge and/or hydride transfer product ions after 12 months compared to the first measurement. We hypothesize three possible factors could be related specifically to the increase in charge and/or hydride transfer product ions over time: (1) the increase in inlet flow rate ($260\,\mathrm{cm^3\,min^{-1}}$ at 0 months to $290\,\mathrm{cm^3\,min^{-1}}$ at $+12$ months), (2) capillary insertion depth, and (3) leaks into the sampling system from maintenance. Lab 6 reports that after maintenance on their instrument, changes in instrument performance (e.g., sensitivity) were observed and may be associated with cleaning the capillary that serves as the inlet to the instrument (Jensen et al., 2023). The instrument was in a stable condition after maintenance before the PIDs were collected. Although we did not observe a strong dependence of $NO^+$ and $O_2^+$ chemistry on capillary insertion distance for the Lab 1 instrument (Fig. S5), it is possible that at the higher inlet flow rates, used for the Lab 6 measurements, an effect could be observed.

None of the product ions from this example change their contribution to the PID by more than 10 % over time – with the exception of the ethanal water cluster. This time-dependent variability in PIDs demonstrated in Fig. 9 points to some factor or combination of factors affecting PIDs not considered in our analyses (e.g., degradation of the microchannel plate detector or possibly ion source degradation; Müller et al., 2014). Additionally, the variability of individual product ions over time provides an estimate of aging variability on the order of 10 % (but as high as 20 %).

### 3.3 Measurements of PIDs for oxygenated VOCs from Lab 1

We highlight features of PID formation from VOCs with oxygenated functionalities that may be measured in high concentrations from samples of indoor air and/or urban air plumes in the sections below. Product ion formation is characterized in the literature for some VOCs like aromatics and monoterpenes (Yuan et al., 2017; Misztal et al., 2012; Materić et al., 2017; Kari et al., 2018) that do not readily form water clusters. Product ion formation from oxygenated VOCs is less well characterized, particularly for water cluster formation.

Figure 10 shows PIDs for select VOCs categorized by functional group as measured from Lab 1 using calibration standards (except for the unidentified monoterpene acetate ester which was measured from a restroom air sample). PIDs

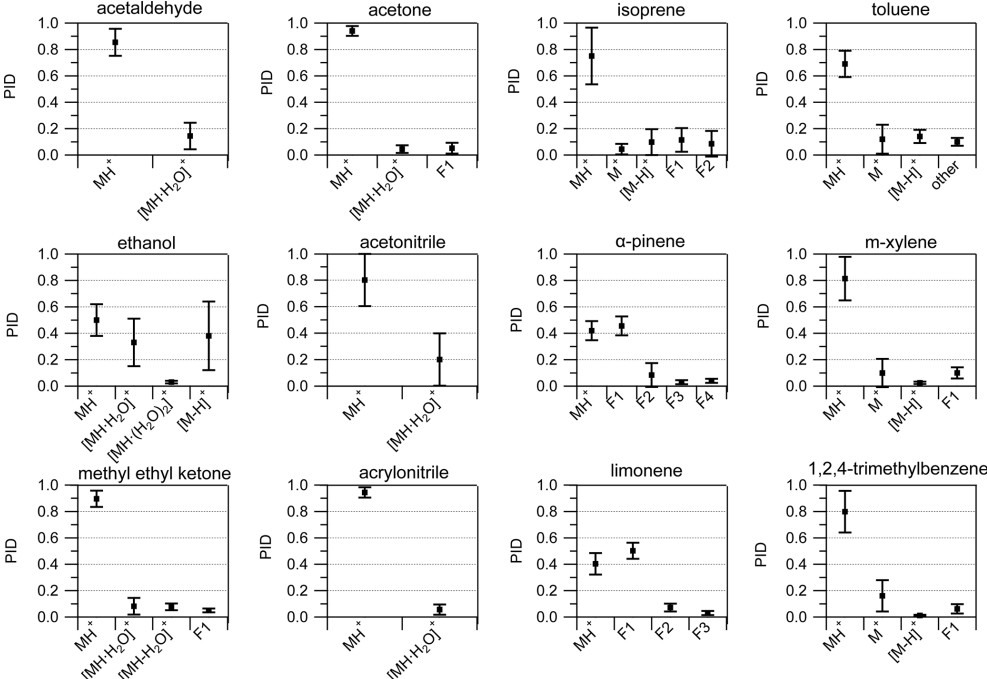

**Figure 8.** Averages (black squares) and standard deviations of the mean ($1\sigma$) of PIDs for select VOCs. Averages were determined from at least five measurements from the interlaboratory comparison dataset. The number of individual measurements used to calculate average and standard deviation values can be found in Table S1 in the Supplement.

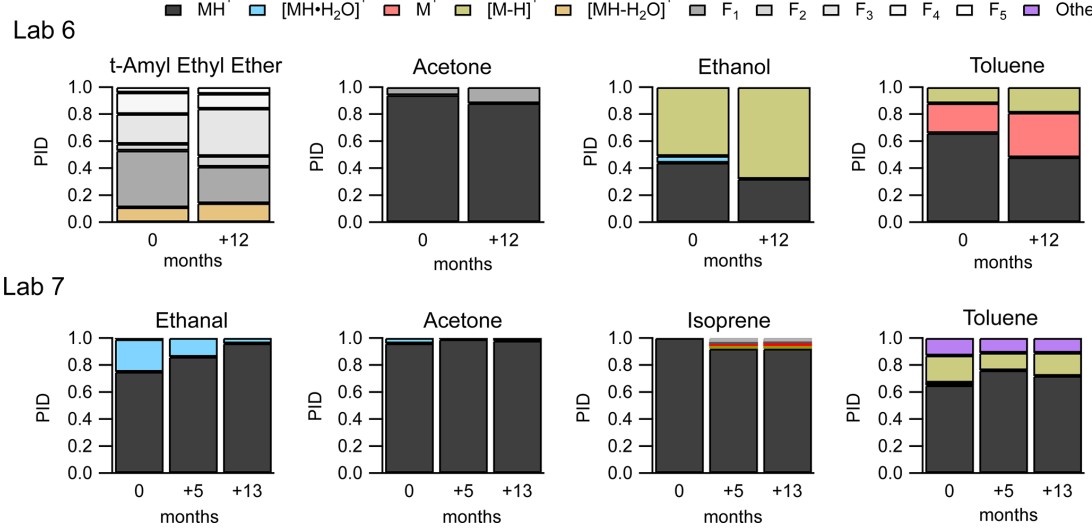

**Figure 9.** PIDs for select VOCs from Lab 6 (top frames) and Lab 7 (bottom frames) showing variability of PIDs over 1 year.

were measured under instrument settings that correspond to Lab 1b in Table 1. A key result demonstrated in Fig. 10 is that, for the subset of VOCs shown here, the H+ adduct contribution to the PID is often less than 60 %, and thus air samples containing these VOCs may have many product ions populating the mass spectra. In other words, $H_3O^+$ ionization (including $NO^+$ and $O_2^+$ impurities) is generating unintended product ions often at rates similar to the intended H+

adduct for most VOCs. Below we discuss general patterns of product ion formation from VOCs with varying functionalities.

### 3.3.1 Saturated aldehydes

Recently, fragment product ions from saturated aldehydes have been highlighted in measurements of urban air influ-

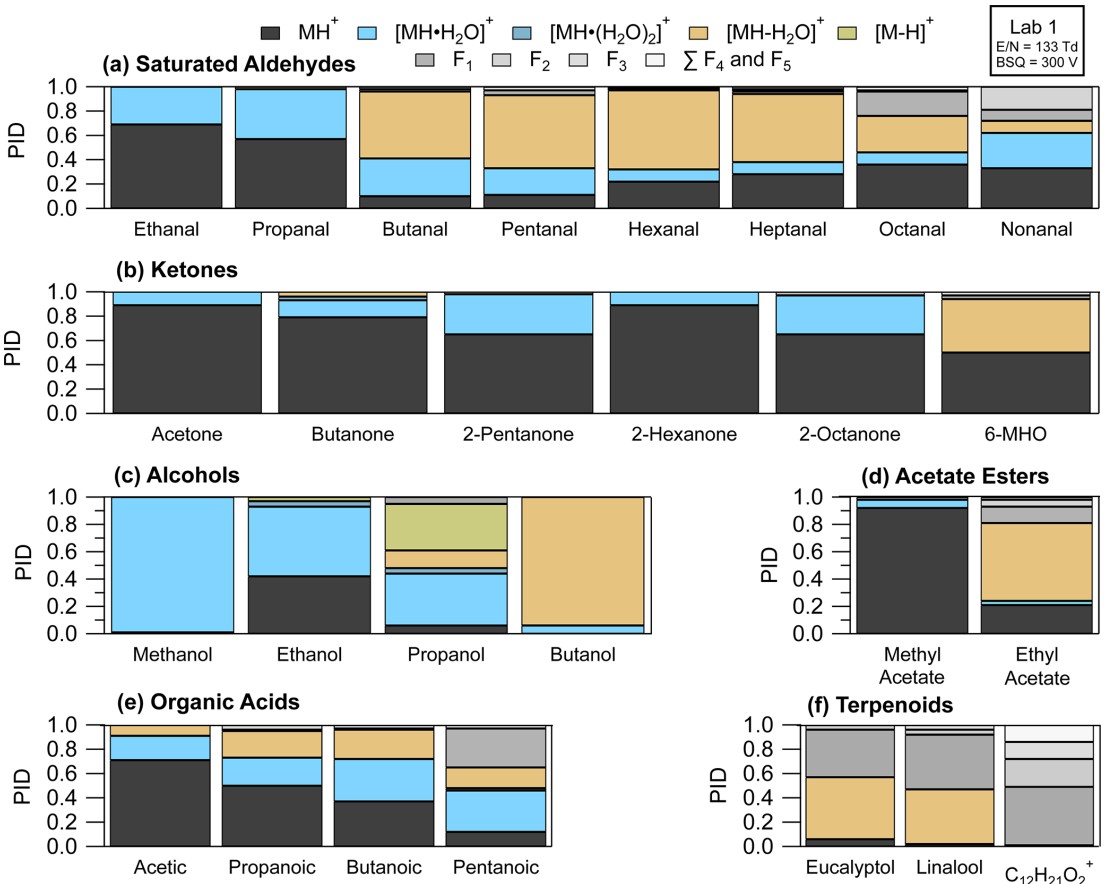

**Figure 10.** PIDs measured for Lab 1 for select VOCs representing different functional groups. VOCs from left to right, per functional group, are in order of increasing carbon number. $C_{12}H_{21}O_2^+$ is an unidentified monoterpene acetate ester, measured from a restroom air sample, likely originating from isobornyl or linalyl acetate (Link et al., 2024).

enced by cooking emissions (Coggon et al., 2024), ozonolysis of seawater (Kilgour et al., 2024), and ozonolysis products of human skin oils in indoor air (Wang et al., 2024; Ernle et al., 2023). In the Lab 1 instrument fragment product ions contributed $> 40\%$ to the PID for saturated aldehydes with a carbon number greater than three (i.e., butanal to nonanal). Water cluster formation contributed $> 20\%$ to the PID for ethanal (acetaldehyde), propanal, and nonanal. As reported previously for butanal through heptanal (Buhr et al., 2002), the fragment ion making the largest contribution to the PID in the Lab 1 instrument was the dehydration product (i.e., $[MH-H_2O]^+$). We find additional agreement with previous literature reporting octanal and nonanal fragmentation to smaller product ions (e.g., $C_5H_9^+$, $C_3H_5^+$, $C_6H_{11}^+$). We suspect, from limited experimental data (Španěl et al., 2002), that larger saturated aldehydes (e.g., decanal) may also produce fragment product ions smaller than the dehydration product ion in the Lab 1 instrument. However, as the carbon number of the saturated aldehyde increases, from butanal, the contribution of the $H^+$ adduct to the PID increases and the contribution of dehydration and fragment product ions decreases, suggesting larger aldehydes fragment less overall than butanal, pentanal, and hexanal. Finally, we note we cannot easily explain the formation of some product ions from $H_3O^+$ ionization from typical mechanisms (e.g., $C_5H_9^+$ from nonanal), and thus we hypothesize that reactions involving $NO^+$ and/or $O_2^+$ may be responsible for the generation of some fragment ions from saturated aldehydes.

### 3.3.2 Ketones

In contrast to saturated aldehydes, and consistent with previous work (Buhr et al., 2002), the saturated ketones (i.e., all the ketones in Fig. 10b except 6-MHO) measured with the Lab 1 instrument do not fragment substantially (i.e., sum of fragment contributions to PID $< 5\%$). However, the saturated ketones do form water clusters with contributions ranging from $10\%$ (e.g., acetone) to $40\%$ (e.g., 2-octanone) to the PID. We do not observe a clear relationship between increasing carbon number and water clustering. In fact, when comparing 6-methyl-5-hepten-2-one (6-MHO) and 2-octanone, two eight-carbon molecules, the water cluster for 2-octanone contributed $40\%$ to the PID, whereas 6-MHO had no de-

tectable water cluster formation (Fig. 10b). Additionally, as demonstrated by the PID from 6-MHO, adding carbon branching and/or additional functionalities can change product ion formation considerably compared to 2-octanone – the saturated $C_8$ ketone analog.

### 3.3.3 Alcohols

We observed important contributions of water clusters ($> 40\%$) to the PIDs measured for methanol, ethanol, and propanol. Methanol and ethanol can be present in concentrations that exceed $1\,\text{nmol}\,\text{mol}^{-1}$ in both outdoor and indoor air (Nazaroff and Weschler, 2024), and thus the water clusters of these two alcohols may make important contributions to sample mass spectra. We also measured small contributions of double water clusters to the PID from ethanol and 2-propanol ($4\%$ for each VOC). Previous studies have shown considerable fragment product ion production from dehydration of alcohols (Buhr et al., 2002; Španěl et al., 2002; Warneke et al., 2003; Pagonis et al., 2019), and we also observed that for 2-propanol and 1-butanol. For 1-butanol $> 90\%$ of the PID was from the dehydration product ion, and we did not measure any generation of the $H^+$ adduct. We also observe small contributions of the hydride transfer product from ethanol that have been reported from another PTR-ToF-MS (Coggon et al., 2024) and measured with the $NO^+$ reagent from a selected-ion flow tube study (Španěl et al., 2002). The hydride transfer product made a $30\%$ contribution to the PID measured for 2-propanol. As summarized in Koss et al. (2016), several other saturated alcohols have hydride transfer enthalpies that decrease with increasing carbon number, and thus hydride transfer product ions may appear in PTR-MS spectra from ambient air samples where saturated alcohols may be highly abundant. As an example, Buhr et al. (2002) measured a $10\%$ contribution of the hydride transfer product from 1-octanol and 2-octanol to their PIDs.

Although we focus on reaction with $NO^+$ as the primary reagent producing hydride transfer products from reaction with VOCs, Hegen et al. (2023) hypothesized that charge transfer from $O_2^+$ to methanol (and possibly other alcohols), with subsequent loss of hydrogen atom, may be an important mechanism for creating product ions that appear in the mass spectrum as hydride transfer products. Thus, both charge and hydride transfer enthalpies may be useful qualitative indicators for predicting if $[M-H]^+$ product ions are generated from ionization of alcohols. For VOCs whose PIDs are not included in the $H_3O^+$ PID library, we refer the reader to Koss et al. (2016) for a table of hydride and charge transfer enthalpies for many VOCs measured using PTR-MS as a useful resource for predicting the possible generation of product ions.

### 3.3.4 Acetate esters, organic acids, and oxygenated monoterpenes

Neither the acetate esters nor oxygenated monoterpenes in this study show a propensity to form water clusters. We measure considerable fragmentation of ethyl acetate (Fig. 10d). In addition to ethyl acetate, Buhr et al. (2002) measured major contributions of fragmentation products of several other acetate esters to their PIDs. Although Buhr et al. (2002) used an older model of PTR-MS with a drift tube ionization region, we expect that larger acetate esters may also fragment to the same degree as observed in that study in the Vocus PTR-ToF-MS.

Alkanoic acids have PIDs that show complexity similar to the saturated aldehydes with extensive water cluster formation and fragmentation (Fig. 10e). Notably, the fraction of the $H^+$ adduct in the PID decreases with increasing carbon number, with roughly $15\%$ of the PID for pentanoic acid allocated to the $H^+$ adduct. More data are needed, but this trend suggests larger organic acids (i.e., $> C_5$) may also produce water cluster and fragment product ions in similar abundance to the $H^+$ adduct. Characterization of PIDs for larger (e.g., $C_9$ and $C_{10}$) acids may be of particular importance for measurements of early generation oxidation products of terpenes.

Notably, the contributions of the $H^+$ adduct to the PID for the terpenoids highlighted here are all less than $5\%$. The monoterpene alcohols (eucalyptol and linalool) generate dehydration product ions with abundances greater than $40\%$ (Fig. 10f). The dehydration product of the monoterpene alcohols, $C_{10}H_{17}^+$, is isobaric (i.e., occurring at the same $m/q$) with the $H^+$ adduct for monoterpenes. We also highlight the PID measured for $C_{12}H_{21}O_2^+$, a monoterpene acetate ester (most likely linalyl or isobornyl acetate based on offline GC analysis presented in Link et al., 2024) measured from a restroom air sample. This ion fragments, losing a neutral acetic acid, to form $C_{10}H_{17}^+$, suggesting monoterpene acetate esters may also create monoterpene interferences from samples where monoterpenes and the acetate esters are both present.

### 3.4 Mass spectral ambiguity from the influence of PIDs: a restroom air sample case study

One consequence of multi-product ion generation in PTR-MS is that if PIDs are unknown or uncharacterized they can create ambiguity when identifying peaks in the mass spectrum in the absence of a pre-separation method. In particular, studies performing non-targeted analysis of the ion signals measured by PTR-MS from indoor air samples (Link et al., 2024; Ditto et al., 2023; Mattila et al., 2021; Liu et al., 2024; Klein et al., 2016) may be challenged by the presence of unintended product ions generated by high concentrations of parent VOCs. For instance, Ernle et al. (2023) recently demonstrated the challenge of quantifying isoprene from $m/q$ 69.07 ($C_5H_9^+$) because of interferences from fragments of aldehydes generated from ozone skin oil oxidation

indoors. We briefly demonstrate several challenges related to product ion generation and resulting mass spectral ambiguity using a measurement of ambient air in a restroom as a case study.

High concentrations of terpenoids emitted from fragrant urinal screens reacted with ozone to create oxidized VOCs in the restroom we sampled from. Figure 11 shows the selected-ion chromatograms for three ions measured, using GC-PTR-ToF-MS, from the restroom air sample to demonstrate challenges associated with product ion formation.

In the restroom the ion possibly attributable to propylene glycol, $C_3H_9O_2^+$ (Hopstock et al., 2024), was found to be mostly comprised of the acetone water cluster. Acetone generates a water cluster with a roughly 10 % efficiency in the Lab 1 instrument used for this restroom measurement. Acetone concentrations are generally elevated indoors compared to outdoors, and in the restroom acetone concentrations were elevated at approximately 20 nmol mol$^{-1}$ (equivalent to 20 parts per billion). Recent studies have used PTR-MS for the measurement of VOCs, including propylene glycol, in the smoke of electronic cigarettes (Bielik et al., 2024; Hopstock et al., 2024; Sheu et al., 2020). Sheu et al. (2020) could not quantify possible contributions of propylene glycol to third-hand smoke indoors because of the acetone water cluster interference. This $C_3H_9O_2^+$ interference from the acetone water cluster may be most pronounced indoors where air can contain elevated acetone concentrations from human breath and material emissions (Molinier et al., 2024).

Acrolein ($C_3H_4O$) is a hazardous indoor air pollutant (Seaman et al., 2007; Logue et al., 2011) and recently was measured, using PTR-MS, from a residential test facility (Arata et al., 2021) where concentrations were high enough such that it was the largest source of gas-phase hazardous exposure (Hodshire et al., 2022). In the restroom the $C_3H_5O^+$ ion signal (i.e., the $H^+$ adduct ion commonly attributed to acrolein) experienced considerable interferences from fragmentation of VOCs containing 9 ($C_9$) to 12 ($C_{12}$) carbon atoms. There were some additional interferences from unidentified sources – one of which may be the propanal hydride transfer product (could not be confirmed here due to coelution of acetone). In the restroom where terpenoid (monoterpenes, monoterpene alcohols, and monoterpene acetate esters) concentrations were roughly 20 nmol mol$^{-1}$, the fragmentation of two ions likely attributable to terpenoids, $C_{10}H_{21}O^+$ and $C_{10}H_{21}O_2^+$, makes important contributions (56 %) to the $C_3H_5O^+$ ion signal. We note that the terpenoids emitted from the urinal screens created high concentrations that may uniquely impact the $C_3H_5O^+$ signal compared to other indoor environments. However, this observation points to the possible unexpected impact of consumer product emissions on indoor air measurements of acrolein.

We highlight here the possible interferences in the $C_{10}H_{17}^+$ ion, normally attributed to monoterpene isomers, from fragmentation reactions of monoterpene alcohols (eucalyptol and linalool) and monoterpene acetate esters (likely isobornyl or linalyl acetate). Previous studies have pointed to $C_{10}H_{17}^+$ interferences from dehydration of monoterpene alcohols of biogenic origin (Joó et al., 2010; Kari et al., 2018; Demarcke et al., 2010). In the restroom we found 25 % of the $C_{10}H_{17}^+$ signal was attributable to dehydration of linalool and eucalyptol, which were emitted from urinal screens. This highlights how in indoor spaces personal care products and scented consumer goods can emit terpenoids (not typically measured in high concentrations from biogenic sources) in high concentrations that can complicate the measurement of monoterpenes using PTR-MS without pre-separation. Additionally, we show a $C_{10}H_{17}^+$ interference from the loss of acetic acid from monoterpene acetate esters, which is possibly a problem unique to the measurement of indoor air.

### 3.5 Using PIDs to improve identification and quantification of VOCs from PTR-MS measurements

#### 3.5.1 Method 1: estimating product ion abundance from real-time data

In Sect. 3.4 we demonstrated the interference of the acetone water cluster in the ion signal, $C_3H_9O_2^+$, that might be typically attributed to propylene glycol (Fig. 11) using a chromatographic pre-separation. If the PID for a given VOC has been measured from a calibration source, then the ratio of product ions can be used to constrain the likely abundance of one product ion relative to another in an ambient air sample measured without chromatographic pre-separation. For example, we can determine the influence of the acetone water cluster on the $C_3H_9O_2^+$ ion signal measured by the PTR-MS, without chromatographic pre-separation (real-time data), by calculating the expected contribution predicted by the acetone PID. We show an example of how we estimated the influence of the acetone water cluster on the real-time $C_3H_9O_2^+$ ion signal in Fig. 12.

We measured the PID for acetone (as shown in Fig. 10 and listed in the $H_3O^+$ PID library) as 0.90 $H^+$ adduct ($C_3H_7O^+$) and 0.10 water cluster ($C_3H_9O_2^+$). Assuming contributions of isomers or product ions to the $C_3H_7O^+$ signal are negligible, we can divide the product ion fraction for $C_3H_9O_2^+$ ($f_{[MH \cdot H_2O]^+}$) by the product ion fraction for $C_3H_7O^+$ ($f_{MH^+}$) to get the fraction of the acetone water cluster relative to the acetone $H^+$ adduct ($\frac{f_{[MH \cdot H_2O]^+}}{f_{MH^+}}$). We can then multiply this fraction by the $C_3H_7O^+$ signal ($S_{MH^+}$) to get the contribution of the acetone water cluster to the $C_3H_9O_2^+$ signal ($S_{[MH \cdot H_2O]^+}$) following Eq. (2),

$$S_{[MH \cdot H_2O]^+} = S_{MH^+} \cdot \frac{f_{[MH \cdot H_2O]^+}}{f_{MH^+}}. \tag{2}$$

Multiplying the $C_3H_7O^+$ signal (shown in Fig. 12a) by $\frac{f_{[MH \cdot H_2O]^+}}{f_{MH^+}}$ (i.e., 0.10/0.90 ≈ 0.11) generates an estimated $C_3H_9O_2^+$ ion signal time series (Fig. 12b, blue trace) that is

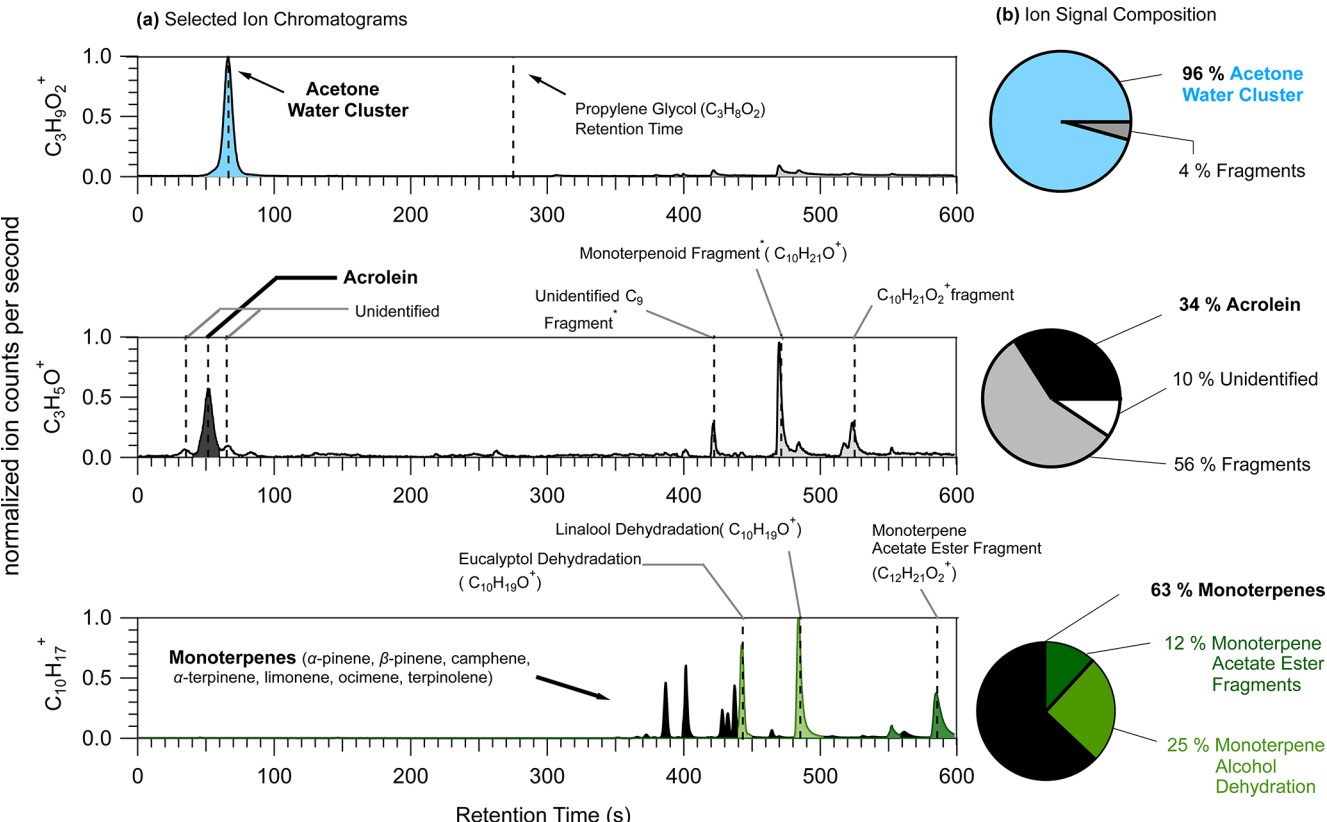

**Figure 11. (a)** Selected-ion chromatograms (left) of three ions for which PIDs present challenges: $C_3H_9O_2^+$ (top), $C_3H_5O^+$ (middle), and $C_{10}H_{17}^+$ (bottom). Dotted vertical lines are placed at the retention times assigned to VOCs or parent ion species either directly measured from calibration sources or supported by time series correlations with known product ions. Peak assignments with an asterisk are species that were assigned from product ion time series analyses. **(b)** Pie charts showing the ion signal composition with contributions from the VOC typically assigned to the ion (black) and contributions from interfering product ions. Product ion contributions to the ion signal are determined by integrating areas of all the major peaks, calculating the relative contribution of each peak to the total area of all the identified peaks, and classifying them by product ion identity.

from the acetone water cluster. In Fig. 12c we calculate the percent residual $C_3H_9O_2^+$ signal after subtracting out the estimated contribution of the acetone water cluster. The average residual of $-0.5\%$ indicates that nearly all of the $C_3H_9O_2^+$ ion signal measured from the restroom is from the acetone water cluster, which is consistent with what we measured from the chromatographic separation in Fig. 11a. Although not shown in this example of $C_3H_9O_2^+$, if after applying this method residual signal remained and was consistently above zero, that could indicate ion signal related to $H^+$ adducts of VOCs or influences of other product ions. We verified that the $C_3H_7O^+$ signal we measured from the restroom (using GC) was $> 95\%$ (with some possible contribution from propanal and contributions of fragment ions) attributable to acetone, thus suggesting that application of this method may work best when supplemented with a GC measurement.

We point to the study of Coggon et al. (2024) for further demonstrations of how to separate the influence of product ions on $H^+$ adduct ions for benzene ($C_6H_7^+$), iso-

prene ($C_5H_9^+$), and ethanal (acetaldehyde, $C_2H_5O^+$) measured from outdoor air influenced by oil and gas and cooking emissions. When directly measuring PIDs using a calibration source is not possible, the $H_3O^+$ PID library included with this paper can serve as a useful source for estimating possible product ion interferences. The existing PTR library compiled by Pagonis et al. (2019) contains measurements of fragment product ions that can also provide product ion data relevant for instruments other than the Vocus. This product ion estimation method may produce reasonable results for some VOCs like acetone, but many ions will often have multiple isomers or isobaric product ion interferences that challenge accurate application of the method.

### 3.5.2 Method 2: using product ions for quantification

PTR-MS quantification is often performed using calibrations of an $H^+$ adduct signal for a target VOC (e.g., $C_3H_7O^+$ for acetone), but the PTR-MS can also be calibrated to product ions. Coggon et al. (2024) showed that benzene con-

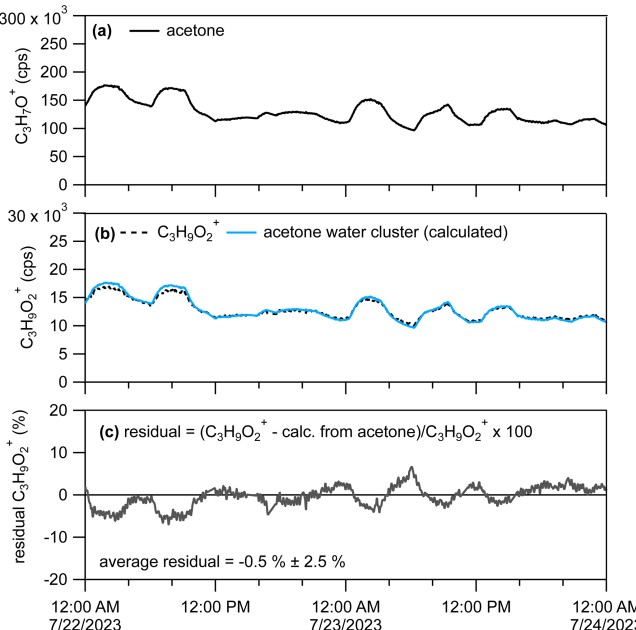

**Figure 12. (a)** Time series for $C_3H_7O^+$ attributable to acetone. **(b)** Time series for $C_3H_9O_2^+$ with the raw signal (dotted black line) and $C_3H_9O_2^+$ calculated to be attributable to the acetone water cluster (10 % contribution to acetone PID). **(c)** Percent residual $C_3H_9O_2^+$ ion signal after subtracting out the estimated contribution from the acetone water cluster.

**Table 3.** Observed and recommended uncertainties for ranges of product ion contributions to a PID for VOCs in the PTR $H_3O^+$ library.

| Product ion fractional contribution to PID range | Single measurement uncertainty | Repeat measurement uncertainty | Recommended reporting uncertainty |
|---|---|---|---|
| > 0.30 | 5 % | 6 % | 15 % |
| 0.16 to 0.30 | 5 % | 10 % | 20 % |
| 0.04 to 0.15 | 11 % | 30 % | 30 % |
| < 0.04 | 50 % | 100 % | 100 % |

### 3.6 The $H_3O^+$ PTR PID library and recommendations for reporting product ion uncertainty

We have compiled the data presented in this paper into a library included in the Supplement. The library will be updated as new observations are included, and the updated library can be found online (Link, 2024). The measurements included in the library were collected under different instrument conditions (listed under the "2_Lab_ID" tab of the library spreadsheet), so care should be taken to most closely compare PIDs reported in this library to PIDs collected on an instrument with a similar configuration (i.e., similar $E/N$, BSQ voltage, ion optic voltages, flow rates). There is an inherent precision with which PIDs can be measured following the GC-based method we have demonstrated. To constrain the uncertainty associated with the PIDs in the $H_3O^+$ PTR PID library, we evaluate the variability in PIDs determined from a single measurement of a VOC (Fig. S6) and the variability observed in PIDs measured from select VOCs over 3 weekends from restroom air samples compared to the PID library measurement performed 6 months earlier (Fig. S7).

We observe that for a single measurement, the contribution of a given product ion to the PID for nonanal varies by no more than 0.01 fractional units (Table S2). For repeat measurements over time (3 weeks for the restroom examples shown here), we observe that the absolute variability in product ion contributions to a PID is largest for product ions with the largest relative contributions to the PID (Table S3). For example, from the restroom samples, the fractional contribution of $C_7H_9^+$ to the toluene PID ranged from 0.71 to 0.78 (a 0.07 fractional unit range) over the 3 weekends, whereas the contribution of $C_6H_7O^+$ ranged from 0.04 to 0.06 (a 0.02 fractional unit range). For both single measurements and the repeat PID measurement example shown in Fig. S7, the relative standard deviation of calculated fractional product ion contributions increases as the absolute contribution decreases.

Thus, we define uncertainty for ranges of product ion fractional contributions to a PID, for a single measurement and repeat measurements performed on the timescale of weeks, as shown in Table 3.

centrations calculated from the charge transfer product ion ($C_6H_6^+$) calibration agreed with concentrations quantified from GC measurements. The authors concluded that the benzene charge transfer product ion ($C_6H_6^+$), which had no interferences, was a more suitable signal to quantify benzene from than the $H^+$ adduct ($C_6H_7^+$), which suffered interferences from fragments of functionalized aromatics. However, pre-separation was used in that study to verify the charge transfer product was free of interferences. In principle, any product ion that is free of interferences could be used as an alternative to the $H^+$ adduct for quantification.

### 3.5.3 Method 3: supplemental measurement with a GC

It is worth acknowledging the value of a supplemental measurement using GC. When directly interfaced to the PTR-MS, GC can be used to measure PIDs and aid in identifying ion signals from the real-time PTR-MS measurement. Benchtop GCs optimized for thermal desorption measurements can also be used in offline analysis to identify possible sources of ion interferences. Although not discussed here, isomers are confounding influences on the interpretation of ion identities, and GC is also useful for quantification of VOC isomers. Nevertheless, not all VOCs present in an air sample are likely to be independently separated (e.g., sesquiterpenes for mid-polarity columns) or trapped and desorbed via a preconcentration system.

The "single-measurement uncertainty" reflects the precision with which the fractional contribution of a given product ion to a PID can be determined from a single measurement. We derived the ranges shown in Table 3 from the calculation of the nonanal PID from a GC measurement. We assume this uncertainty is not chemical dependent and thus applies to other chemicals. The single-measurement uncertainty values are a conservative estimate of the uncertainty associated with the calculation of a product ion contribution to a PID when measured using the GC method.

The "repeat-measurement uncertainty" reflects the precision of a product ion's fractional contribution to a PID when repeatedly measured over the timescale of weeks (supported by the measurements from the restroom shown in Fig. S7). We used the variability in product ion contributions calculated for the acetic acid, acetone, and toluene PIDs shown in Fig. S7 and in Table S3 to constrain the repeat-measurement uncertainty. We find that the relative standard deviation from repeat measurements of product ion contributions over 3 weeks is greater than that of a single measurement (Table 3).

We derive a recommended reporting uncertainty by comparing the average and standard deviations of the product ion contributions to the PIDs for acetic acid, acetone, and toluene – measured in the restroom samples – to their corresponding entries in the $H_3O^+$ PTR-MS PID library. The PID measurements presented in the library (for Lab 1b) were acquired approximately 6 months prior to the restroom measurements. Thus, the recommended reporting uncertainty provided in Table 3 incorporates our constraints on repeated measurement uncertainty as well as an estimate of the stochastic variability in PID development that can occur over months as is demonstrated earlier in Fig. 9. By applying the recommended reporting uncertainties to the average product ion contributions measured for the PIDs of the three VOCs in the restroom samples, we find that the average restroom values come into the range of the values in the PID library (Table S3).

### 3.7 Recommendations for mitigating challenges from unintended product ion generation

As demonstrated in the interlaboratory comparison data, PTR-MS users are likely to experience unintended product ion generation under a variety of instrument operating conditions. We recommend several practices that PTR-MS users can adopt to improve the interpretability of PTR-MS data.

– *Measure PIDs regularly.* Surrogate analytes can be used (and included in calibration source cylinders) to provide some indication of how likely it is a mass spectrum may be influenced by certain types of product ions. For example, benzene can be used as a surrogate for charge transfer reaction chemistry, acrolein (data shown in the $H_3O^+$ PTR PID library) for water clustering, and $\alpha$-pinene for fragmentation. Because PIDs can change

over time, regularly (at least once a month during periods of active measurements) measuring the PIDs of a few key surrogates can provide relative information on how the PIDs of other VOCs may also be changing. The ion chemistry presented in Table 1 can act as a guide for users to evaluate if ions appearing in a mass spectrum could be generated from unintended product ions. Additionally, the step-by-step procedure outlined in the Supplement can serve as a method for measuring PIDs.

– *Optimize analyte detection with instrument tuning.* Here we demonstrated IMR $E/N$ and BSQ voltage affected PIDs. A user can measure the PID of target analytes and scan $E/N$ and BSQ voltage values to optimize the production of a desired product ion (e.g., the $H^+$ adduct). Because cluster and fragmentation product ions are generated and detected more efficiently at different extremes of $E/N$ and BSQ voltage values, instrument tuning will not eliminate unintended product ion generation.

– *Refer to the $H_3O^+$ PTR PID library.* For the VOCs available in the library (Link, 2024) a user can identify the problematic $m/q$ and elemental formula associated with unintended product ions from VOCs known to be in a sample (including multi-component calibration sources).

– *Measure the instrument sample flow rate regularly.* We provide evidence suggesting an influence of flow rate on PIDs, but we also note that the sample flow rate will also affect instrument sensitivity (Jensen et al., 2023). When sampling from pristine environments, measuring the sample flow once a week may be sufficient. For measurements of urban or indoor air, measuring the flow once a day is recommended. Higher-frequency flow checks may be necessary for measurements where particulate matter loading is high (e.g., fire research laboratory burn samples, cooking emissions).

– *If possible, use a supplemental measurement, GC or otherwise, to support identification of ions measured with PTR-MS from multi-component air samples.*

– *Define the acceptable level of accuracy for your measurement.* PTR-MS provides high-time-resolution measurements of VOCs in air that cannot be achieved with many techniques. For non-targeted analyses, identifying and accounting for all influences of unintended product ions is currently impractical. Studies that seek to quantify all VOCs measured, both known and unknown, by the PTR-MS may suffer from greater uncertainties arising from unintended product ion generation. While more uncertain, these non-targeted analyses are important for progressing research. On the other hand, users seeking to quantify specific VOCs (e.g., air toxics or

hazardous air pollutants) for the purposes of measurements supporting regulations will need to account for product ion chemistry for high-accuracy measurements.

## 4 Outlook

All reagent ions used for chemical ionization mass spectrometry create unintended product ions that can present challenges when identifying and quantifying VOCs. Continued work characterizing and constraining the impact of instrument operating parameters and sampling methods on product ion generation is warranted to leverage the sensitivity, selectivity, and versatile sampling capabilities that field-deployable chemical ionization mass spectrometers provide. PTR-MS users should be aware that product ion generation (of not only fragments but also charge and/or hydride transfer and water clusters) occurs for most VOCs to varying degrees. Additionally, the ambiguity created from product ion contributions to mass spectra measured from chemically complex samples may create challenges to accurate identification and quantification of VOCs – particularly for non-targeted analyses. Further characterization of PIDs across many PTR-MS instruments may be useful in constraining interferences and decreasing the uncertainty from their influence on mass spectra.

There is a current interest in developing standardized methods of measurement using chemical ionization mass spectrometers. Currently, no standard methods for sampling with PTR-MS or other chemical ionization instruments exist. Notable research efforts towards standardization methods of PTR-MS measurements include the development of ion libraries (Pagonis et al., 2019; Yáñez-Serrano et al., 2021), calibrations and standard reference materials (Worton et al., 2023; Jensen et al., 2023; Sekimoto et al., 2017), data analysis methods (Holzinger, 2015; Cubison and Jimenez, 2015), and interlaboratory comparison studies (Holzinger et al., 2019). Continued efforts, particularly in the form of coordinated interlaboratory comparison studies, would be useful for the development of standard operational procedures and practices.

## 5 Summary and conclusions

Here we outlined general rules for identifying possible product ion interferences based on common reaction mechanisms that can occur when using PTR-MS. Additionally, the method of product ion classification (using the ion formula predicted from mechanisms) used here can be employed in future studies to continue to develop product ion libraries using a consistent methodology so that PIDs can be compared directly from different studies. Consistent with the decades of previous research, which includes measurements on PTR-MS instruments that use a drift tube for ionization, we observe $E/N$ as a predictor of the extent to which clustering or

fragmentation product ions contribute to the PID of a VOC. Of particular importance for the instruments in this study is also the influence of $\Delta V_2$ in creating $E/N$-like effects on PIDs and the BSQ RF voltage affecting PIDs through mass discrimination.

We demonstrate here that instrument tuning can affect PIDs, but tuning can also affect instrument sensitivity. We do not discuss the relationship between instrument tuning, product ion formation, and instrument sensitivity here but instead point the reader to Li et al. (2024) for a detailed evaluation of this relationship relevant for Vocus PTR-ToF-MS instruments. However, we note that specific instrument tuning properties explored here have implications for instrument sensitivity. For instance, Li et al. (2024) showed that the H⁺ adduct contribution to the PID and sensitivity for 1,3,5-trimethylbenzene did not change appreciably with increasing $E/N$, whereas the H⁺ adduct contribution to the PID and sensitivity for hexanal (PID shown in Fig. 10) decreased with increasing $E/N$. This comparison demonstrates that VOCs susceptible to fragment ion formation may show decreasing sensitivity to the H⁺ adduct with increasing $E/N$. In addition to $E/N$, we show that as the voltage difference between the BSQ front and skimmer ($\Delta V_2$) increases, this can increase fragmentation and decrease water clustering product ion contributions to the PIDs (Fig. 5).

In another example, we demonstrated that higher BSQ voltages can filter out lower $m/q$ ions and affect measured PIDs, but another implication of higher BSQ voltages is that the sensitivity of the H⁺ adduct for lower-molecular-weight species (e.g., formaldehyde, acetonitrile, formic acid) will also decrease. Interlaboratory comparisons focusing on constraining the relationship between PIDs and instrument sensitivity would be informative for the development of standard tuning configurations optimized for the measurement of specific VOCs or types of VOCs (e.g., aldehydes, aromatics).

Despite having similar operating conditions (i.e., similar $E/N$ and BSQ voltage settings), PIDs measured across laboratories showed considerable variability. Further, PIDs measured from the same instrument over time were not consistent. Our observations support the conclusion that if a user configures the same model PTR-MS identically to an instrument in the literature, they should not expect identical PIDs. Additionally, a user may expect different PIDs from the same instrument after several months.

However, we also show that some of the variability in PIDs between instruments was explainable from qualitative arguments. For example, Lab 6 operated with the highest $E/N$ and showed the largest contributions of fragmentation and charge and/or hydride transfer products to PIDs and small contributions from water clusters compared to the other labs. Qualitative arguments based on $E/N$ or BSQ voltage could not completely explain the variation in water clustering between labs. The quantitative constraints on PIDs presented here could be improved with continued input of data from users to the H₃O⁺ PID library (included here as a supple-

mental document). Future work from our group at NIST will focus on integrating measurements of PIDs contained in the existing PTR library from Pagonis et al. (2019) with the $H_3O^+$ PID library included here. We encourage users to continue to contribute data for inclusion in the $H_3O^+$ PID library in continued efforts to understand PIDs and standardize methods of PTR-MS measurements.

*Data availability.* Additional analyses of instrument configuration on PIDs are presented in the Supplement. A spreadsheet containing the PID data from the interlaboratory comparison (the $H_3O^+$ PID library) is included in the Supplement, and the most up-to-date versions can be retrieved online from https://doi.org/10.18434/mds2-3582 (Link, 2024). Users wishing to submit data to this library can email the corresponding author (michael.f.link@nist.gov), and a link to submit a data file will be provided. More details can be found in the "ReadMe" tab of the supplemental $H_3O^+$ PID library.

*Supplement.* The supplement related to this article is available online at https://doi.org/10.5194/amt-18-1-2025-supplement.

*Author contributions.* MFL, MSC, AAA, DK, PAH, MC, CES, AJ, JY, HNH, JCD, CW, WD, KG, SJ, RLR, JG, TB, JPDA, NBD, and DP collected measurements of calibration gas and/or sample gas on PTR-MS instruments and provided data for analysis. MFL, DP, MSC, JCD, HNH, AJ, AAA, PAH, KG, NBD, and MC analyzed and organized data. Conceptualization and writing of the manuscript was performed by MFL, DP, MSC, MC, NBD, JCD, and AJ. All authors reviewed and provided feedback on drafts of the manuscript.

*Competing interests.* The contact author has declared that none of the authors has any competing interests.

*Disclaimer.* Certain equipment, instruments, software, or materials, commercial or non-commercial, are identified in this paper in order to specify the experimental procedure adequately. Such identification is not intended to imply recommendation or endorsement of any product or service by NIST, nor is it intended to imply that the materials or equipment identified are necessarily the best available for the purpose.

Publisher's note: Copernicus Publications remains neutral with regard to jurisdictional claims made in the text, published maps, institutional affiliations, or any other geographical representation in this paper. While Copernicus Publications makes every effort to include appropriate place names, the final responsibility lies with the authors.

*Acknowledgements.* We would like to acknowledge the National Research Council Research Associateship Program; the Alfred P. Sloan Foundation; and the U.S. Department of Energy (DOE), Office of Science, Office of Biological and Environmental Research, Atmospheric System Research (ASR).

*Financial support.* This research has been supported by the National Research Council Research Associateship Program and Alfred P. Sloan Foundation (grant no. G-2019-11404) and partially supported by the DOE, Office of Science, Office of Biological and Environmental Research, ASR (award no. DE-SC0021985).

*Review statement.* This paper was edited by Hendrik Fuchs and reviewed by two anonymous referees.

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
