# Peer review of "Product Ion Distributions using $\text{H}_3\text{O}^+$ PTR-ToF-MS: Mechanisms,"

_EGUsphere, 2024_

## Referee Comment (RC1)

**General comments**

The manuscript titled "Product Ion Distributions using $H_3O^+$ PTR-ToF-MS: Mechanisms, Transmission Effects, and Instrument-to-Instrument Variability" by Link et al. explores how ion-molecule reactor conditions, ion transmission effects from quadrupole and ion optical tuning, and inlet capillary configuration influence the measured product ion distributions in $H_3O^+$ PTR-ToF-MS. Their manuscript establishes a library of $H_3O^+$ product ion distributions for nearly 100 VOCs and provides several practical recommendations to improve the interpretability of PTR-MS data. The research goal is novel and holds significant application relevance for the mass spectrometry community. I recommend its publication in AMT after addressing several minor comments.

**Minor comments**

1. Abstract Conciseness: The abstract is overly lengthy, some of the introductory content regarding PTR-MS could be moved to the introduction. Please condense the abstract.

2. Line 203-206: The earlier description has already clearly outlined the advantages of GC separation. Therefore, the statement "Though all the PIDs we present here were determined from GC-PTR-ToF-MS measurements, PIDs can be determined without pre-separation from single component calibration sources. Without pre-separation, multicomponent VOC sources may create product ions that can interfere with quantification of the PIDs from a given VOC." appears redundant.

3. In Section 3.1, the authors illustrate the influence of instrument configuration on PIDs using examples such as pentanoic acid, ethanol, and toluene. Have the authors considered whether the PIDs of different functional group types exhibit consistent responses to instrument parameters, or do the responses vary uniquely for different species?

4. In Section 3.2, the authors discuss the uncertainties in PIDs resulting from variations across different laboratories and over time. It would be beneficial to provide an overall uncertainty estimate to enhance the applicability of the $H_3O^+$ PID Library.

5. Humidity may affect PIDs. Were all samples measured under the same humidity conditions? Has the potential impact of humidity been considered?

6. In the $H_3O^+$_PID_Library: In lab1b, the $C_{10}H_{17}^+$ for linalool should perhaps be $[M-H_2O]^+$ rather than listed in column F1. Please carefully review the table to avoid similar errors.

---

## Author Comment (AC1)

Response to Reviewers for **Product Ion Distributions using $H_3O^+$ PTR-ToF-MS: Mechanisms, Transmission Effects, and Instrument-to-Instrument Variability**

We thank the reviewers for taking the time to review our manuscript. The reviewers provided comments that helped us recognize several areas in the manuscript where we could improve our communication and provide clarification on our methods and analysis. We included an updated analysis of the ion optic voltage difference impacts on PIDs of several other VOCs (in addition to pentanoic acid). Through this new analysis we found that ion optic voltage differences, representative of the range of voltage differences observed between the labs in the intercomparison, could increase fragmentation of some VOCs (nonanal and 6-MHO in our set of VOCs) to the same extent as equivalent changes in E/N of several Td. We also expanded the discussion on how the effects of each ion transfer component we analyzed individually can compound to change the initial PID that is formed in the IMR into a different PID that is measured. Finally, we included a short discussion (including a new figure and table) on our estimated uncertainties associated with the PID values presented in the $H_3O^+$ PTR PID Library.

Below we show the original reviewer comments in black and our responses in blue.

**Reviewer #1:**

**Comment 1**: Abstract Conciseness: The abstract is overly lengthy, some of the introductory content regarding PTR-MS could be moved to the introduction. Please condense the abstract.

We have condensed the abstract.

"Proton-transfer-reaction mass spectrometry (PTR-MS) using hydronium ion ($H_3O^+$) ionization is widely used for the measurement of volatile organic compounds (VOCs) both indoors and outdoors. $H_3O^+$ ionization, and associated chemistry in an ion molecule reactor, is known to generate product ion distributions (PIDs) that include other product ions besides the proton-transfer product. We present a method, using gas-chromatography pre-separation, for quantifying PIDs from PTR-MS measurements of nearly 100 VOCs of different functional types including alcohols, ketones, aldehydes, acids, aromatics, organohalides, and alkenes. We characterize instrument configuration effects on PIDs and find that reactor reduced electric field strength (E/N), ion optic voltage gradients, and quadrupole settings have the strongest impact on measured PIDs. Through an interlaboratory comparison of PIDs measured from calibration cylinders we characterized the variability of PID production from the same model of PTR-MS across seven participating laboratories. Product ion variability was generally smaller (e.g., < 20 %) for ions with larger contributions to the PIDs (e.g., > 0.30), but less predictable for product ions formed through $O_2^+$ and $NO^+$ reactions. We present a publicly available library of $H_3O^+$ PTR-MS PIDs that will be updated periodically with user-provided data for the continued investigation into instrument-to-instrument variability of PIDs."

**Comment 2**: Line 203-206: The earlier description has already clearly outlined the advantages of GC separation. Therefore, the statement "Though all the PIDs we present here were determined

from GC-PTR-ToF-MS measurements, PIDs can be determined without pre-separation from single component calibration sources. Without preseparation, multicomponent VOC sources may create product ions that can interfere with quantification of the PIDs from a given VOC." appears redundant.

We have removed the statement highlighted by the reviewer.

**Comment 3**: In Section 3.1, the authors illustrate the influence of instrument configuration on PIDs using examples such as pentanoic acid, ethanol, and toluene. Have the authors considered whether the PIDs of different functional group types exhibit consistent responses to instrument parameters, or do the responses vary uniquely for different species?

We chose to highlight pentanoic acid, toluene, and ethanol in Fig. 3 because each VOC generates product ions associated with different chemical mechanisms. The rates of the different reactions that generate different product ions may be affected by E/N. Pentanoic acid generates a diverse PID containing water clusters and fragments. Toluene generates hydride and charge transfer product ions, and ethanol generates a water cluster and hydride transfer product. In our edited version of the manuscript, we have included figures (Fig. 5, Fig. 6, and Fig. S4) that show PIDs for acetone, nonanal, chlorobenzene, and 6-methyl-5-hepten-2-one that vary as a function of the BSQ front – skimmer ion optic relationship. We use these surrogates to describe how the instrument component may affect a PID—whether it's through ion chemistry in the IMR, collisional declustering in the ion optics, or mass filtering with the BSQ.

To evaluate if PIDs predictably vary in response to changes in instrument components, as a function of VOC functional group, would require more measurements and considerable analysis and could be the focus of future work. Nevertheless, in Fig. 10, we show the PIDs measured for VOCs spanning a range of functional groups under a reference set of instrument conditions. When combined with the results presented in Section 3 the reader will get a sense of how different functional groups may respond to changes in instrument components.

We have included the edited text for the manuscript below.

[revised manuscript text omitted]

**Comment 4**: In Section 3.2, the authors discuss the uncertainties in PIDs resulting from variations across different laboratories and over time. It would be beneficial to provide an overall uncertainty estimate to enhance the applicability of the H3O+ PID Library.

We thank the reviewer for this helpful suggestion. Section 3.6 "The $H_3O^+$ PTR PID Library and Recommendations for Product Ion Uncertainty" is a new addition to the manuscript that contains several paragraphs of text, a new figure, and a new table. We use this section to investigate (1) the precision with which we can measure PID contributions from a single measurement ("single measurement uncertainty") and (2) the variability observed from measurements of the PIDs of three select VOCs from restroom air samples that occurred over three weekends ("repeated measurement uncertainty"). By comparing our observations of product ion contribution variability measured from the restroom samples to the product ion contribution values in the $H_3O^+$ PTR-MS PID Library we estimate a "recommended reporting uncertainty" for product ion contributions to a PID. These "recommended reporting uncertainties" now apply to the product ion entries in the PID library. There is also an additional figure in the supplemental information (Fig. S6) and two additional tables (Tables S2 and S3).

Below is the new material included in the manuscript:

**3.6 The $H_3O^+$ PTR PID Library and Recommendations for Reporting Product Ion Uncertainty**

We have compiled the data presented in this manuscript into a library included in the supplement. The library will be updated as new observations are included and the updated library can be found online (NIST, 2024). The measurements included in the library were collected under different instrument conditions (listed under the "2_Lab_ID" tab of the library spreadsheet) so care should be taken to most closely compare PIDs reported in this library to PIDs collected on an instrument with a similar configuration (i.e., similar E/N, BSQ voltage, ion optic voltages, flowrates). There is an inherent precision with which PIDs can be measured following the GC-based method we have demonstrated. To constrain the uncertainty associated with the PIDs in the $H_3O^+$ PTR PID Library, we evaluate the variability in PIDs determined from a single measurement of a VOC (Fig. S6) and the variability observed in PIDs measured from select VOCs over three weekends from restroom air samples compared to the PID library measurement performed six months earlier (Fig. S7).

[Figure]

**Figure S7: PIDs for three VOCs measured by Lab 1 from calibration sources and included in the H₃O⁺ PTR PID Library (right barplots). PIDs for those same VOCs measured over three weekends from restroom air samples are shown in the barplots on the right. The label "Restroom Wk1" indicates the sample that was acquired from the restroom on the first weekend in the measurement set (Wk2 is the second weekend and Wk3 is the third weekend).**

We observe that for a single measurement, the contribution of a given product ion to the PID for nonanal varies by no more than 0.01 fractional units (Table S2). For repeat measurements over time (three weeks for the restroom examples shown here), we observe that the absolute variability in product ion contributions to a PID is largest for product ions with the largest relative contributions to the PID (Table S3). For example, from the restroom samples, the fractional contribution of $C_7H_9^+$ to the toluene PID ranged from 0.71 to 0.78 (a 0.07 fractional unit range) over the three weekends whereas the contribution of $C_6H_7O^+$ ranged from 0.04 to 0.06 (a 0.02 fractional unit range). For both single measurements and the repeat PID measurement example shown in Fig. S7, the relative standard deviation of calculated fractional product ion contributions increases as the absolute contribution decreases.

Thus, we define uncertainty to ranges of product ion fractional contributions to a PID, for a single measurement and repeat measurements performed on the timescale of weeks, as shown in Table 3.

**Table 3: Observed and Recommended Uncertainties for Ranges of Product Ion Contributions to a PID for VOCs in the PTR H₃O⁺ Library.**

| Product Ion Fractional Contribution to PID Range | Single Measurement Uncertainty | Repeat Measurement Uncertainty | Recommended Reporting Uncertainty |
|---|---|---|---|
| > 0.30 | 5 % | 6 % | 15 % |
| 0.16 to 0.30 | 5 % | 10 % | 20 % |
| 0.04 to 0.15 | 11 % | 30 % | 30 % |
| < 0.04 | 50 % | 100 % | 100 % |

The "single measurement uncertainty" reflects the precision with which the fractional contribution of a given product ion to a PID can be determined from a single measurement. We derived the ranges shown in Table 3 from the calculation of the nonanal PID from a GC measurement. We assume this uncertainty is not chemical dependent and thus applies to other chemicals. The "single measurement uncertainty" values are a conservative estimate of the uncertainty associated with the calculation of a product ion contribution to a PID when measured using the GC method.

The "repeat measurement uncertainty" reflects the precision of a product ions fractional contribution to a PID when repeatedly measured over the timescale of weeks (supported by the measurements from the restroom shown in Fig. S7). We used the variability in product ion contributions calculated for the acetic acid, acetone, and toluene PIDs shown in Fig. S7 and in Table S3 to constrain the "repeat measurement uncertainty". We find that the relative standard deviation from repeat measurements of product ion contributions over three weeks is greater than that of a single measurement (Table 3).

We derive a recommended reporting uncertainty by comparing the average and standard deviations of the product ion contributions to the PIDs for acetic acid, acetone, and toluene—measured in the restroom samples—to their corresponding entries in the H₃O⁺ PTR-MS PID Library. The PID measurements presented in the library (for Lab 1b) were acquired approximately six months prior to the restroom measurements. Thus, the recommended reporting uncertainty provided in Table 3 incorporates our constraints on "repeated measurement uncertainty" as well as an estimate of the stochastic variability in PID development that can occur over months as is demonstrated earlier in Fig. 9. By applying the recommended reporting uncertainties to the average product ion contributions measured for the PIDs of the three VOCs

in the restroom samples, we find that the average restroom values come into range of the values in the PID library (Table S3).

**Comment 5**: Humidity may affect PIDs. Were all samples measured under the same humidity conditions? Has the potential impact of humidity been considered?

Several studies (Krechmer et al., 2018; Li et al., 2024a; Zang and Willis, 2024) have found that many VOCs measured by the Vocus PTR-MS do not exhibit a strong sensitivity dependence on sample relative humidity. For instance, the study of Li et al, 2024 observed a maximum increase in sensitivity of < 10 % (at a sample relative humidity of 85 %) for the all the VOCs they measured with the exception of long chain aldehydes which showed an approximate increase in sensitivity of between 24 % to 32 % for butanal and pentanal. We could investigate the influence of relative humidity on PIDs in future work.

For the current work, all of the PIDs we have included in the manuscript and in the library were measured at the same relative humidity (i.e., measured in dry air). For other instruments that show important VOC sensitivity dependencies on humidity the dependency of humidity on PIDs would have to be evaluated.

**Comment 6**: In the H3O+_PID_Library: In lab1b, the C10H17+ for linalool should perhaps be [MH2O]+ rather than listed in column F1. Please carefully review the table to avoid similar errors.

We thank the reviewer for looking at the library. In addition to this error, we caught several additional typos and have revised the library. We will upload this revised version with the final manuscript.

**Reviewer #2:**

**Comment 1**: I am curious if replicates were performed and if PIDs are presented as averages from multiple laboratory tests. Were any approaches taken to ensure these results are robust and statistically sound? Are the differences between measurements and between laboratories greater than the statistical noise? I encourage the authors to present a measure of the precision of the PIDs by collecting multiple measurements over a short time period for 1-2 instruments.

We thank the reviewer for this helpful suggestion. This comment and comment #4 from Reviewer 1 motivated us to constrain the uncertainties reported for the PIDs in this manuscript to a greater extent. We have included the brief response to Reviewer 1 comment #4 below, but the updated manuscript text, figure, and table can be found in the response to this comment above.

Section 3.6 "The $H_3O^+$ PTR PID Library and Recommendations for Product Ion Uncertainty" is a new addition to the manuscript that contains several paragraphs of text, a new figure, and a new table. We use this section to investigate (1) the precision with which we can measure PID contributions from a single measurement ("single measurement uncertainty") and (2) the variability observed from measurements of the PIDs of three select VOCs from restroom air samples that occurred over three weekends ("repeated measurement uncertainty"). By comparing

our observations of product ion contribution variability measured from the restroom samples to the product ion contribution values in the $H_3O^+$ PTR-MS PID Library we estimate a "recommended reporting uncertainty" for product ion contributions to a PID. These "recommended reporting uncertainties" now apply the product ion entries in the PID library. There is also an additional figure in the supplemental information (Fig. S6) and two additional tables (Tables S2 and S3).

**Comment 2**: Title: I was surprised by the simultaneous use of "H3O+" and "PTR" in the title. Is there any proton transfer reaction-MS technique that does not use H3O+?

There are a lot of different chemicals that have been studied that act as Brønsted acids in the gas-phase. Protonated methane was used in most early chemical ionization research, but other smaller hydrocarbons were used after that (Field, 1968; Futrell et al., 1970; Li et al., 2024b). Reviewers of early papers from Mudson and Field were very skeptical of CIMS. Recently, I have talked to some researchers using protonated urea as a CIMS reagent for proton-transfer. To be precise we have specifically noted our focus on $H_3O^+$ in the current study.

**Comment 3**: Line 122: It should probably be mentioned that there is no good control of the ratio of H3O+ vs. NO+ or O2+ primary ions because of the BSQ-induced cutoff. (In traditional PTR-MS instruments, the H3O+ vs. O2+ or NO+ ratio could be optimized using the source valve which the Vocus does not have.)

We agree with the reviewer that there is no good control of the relative amount of primary to secondary reagent ions in the Vocus instruments and have included text in the manuscript to address this. However, we note an important distinction between the amount of secondary product ions generated near the IMR and the amount of secondary product ions that transmit through the instrument and ultimately are measured. In the Vocus there is no good control on the exact production of $NO^+$ and $O_2^+$ because there isn't a good control on the amount of air being ionized in the plasma along with water vapor. The role of the BSQ is to attenuate the transmission of reagent ions after they are generated and have performed chemistry in the IMR. Thus, optimization of the reagent ion distribution that affects PIDs created in the IMR is not possible to our knowledge, but the transmission of measured reagent ions can be affected by changing the BSQ voltage as the reviewer notes.

(Line 120) "We note that in the Vocus instruments used in this study the ratio of $NO^+$ and $O_2^+$ to $H_3O^+$ reagent ions cannot be precisely controlled."

**Comment 4**: Section 2.2.1: How old were the calibration cylinders from the different labs? Were they still certified or could aged, degraded calibration gas cause part of the observed differences? Did all labs apply dead-time correction / transmission efficiency in the same way?

All of the calibration cylinders used in this study were from Apel-Riemer and were certified up to one year after purchase. All cylinder measurements were performed from were less than two years old. This is a good question as stability of VOCs in cylinders is not as well understood as other chemicals like greenhouse gases. We do note that many of the measurements included in

the PTR-MS PID Library from Lab 1 were obtained from evaporated liquid solutions. We added a line of text to the manuscript:

(Line 151) "All calibration gas cylinders were less than two years old."

That being said, we don't think that degradation of VOCs in the cylinders could explain some of the variability in PIDs observed between labs. In order for breakdown products in the cylinders to affect the quantification of PIDs they would have to coelute with the primary analyte. While not impossible, we did not see any evidence in the data of calibration gas decomposition products coeluting with the primary analytes.

All labs sent the raw data to NIST to be processed. The primary author did not apply a transmission correction (ToF duty cycle or otherwise) to the data. Although methods for determining the transmission efficiency exist in the literature, for future work we hope to test different methods for measuring the transmission. We then hope to relate PIDs and sensitivities that would be corrected for transmission. That was beyond the scope of the current work. We added a statement to the main text.

(Line 310) "Data were not ToF duty cycle corrected."

**Comment 5**: Line 252 ff: Increasing the entire set of voltages is not what happens when users use the automatic tuning software "Thuner", which individually changes BSQ or skimmer voltages to increase sensitivity. As a result, unwanted increases in fragmentation can be induced that the users do not even realize if they focus on using Thuner to increase sensitivity. So, at least a statement of caution is warranted. Did all the labs in the intercomparison use this approach of increasing the entire set of voltages simultaneously? If not, I wonder if some of the variability between labs (e.g., variability discussed in Line 411-414) could be a result of tuning these voltages. (Refer to Fig. S2 in Coggon et al. 2024.) If not all labs in the intercomparison used the Brophy & Farmer method for adapting skimmer and BSQ front/back voltages, please report which ones did, and how others decided on the deltaV1 and deltaV2s they used. Some Vocus instruments have two skimmer voltages, and the delta between both might impact the PID. For those instruments where two skimmer voltages exist, reporting the delta between both voltages would be helpful.

Based on the reviewer's comment we recognize a need for clarification with our methods. We have included two statements shown below in the main text.

(Line 264) "We performed these ensemble voltage changes manually without the use of tuning software."

(Line 268) "We performed these PID sensitivity tests to instrument configuration only on the instrument corresponding to Lab 1."

We did not use Thuner to change voltages when performing these sensitivity tests. We would prefer not to mention Thuner, or other tuning software, as it is not part of our analysis. Additionally, these tests of E/N, ion optic voltage, and BSQ voltage were only performed on the

Lab 1 instrument. Processing a similar dataset from several other instruments would be considerable additional work that would likely warrant it's own paper. We added text to clarify which skimmer we evaluated in our tests:

(Line 269) "The skimmer component in the $\Delta V_1$ and $\Delta V_2$ relationships described here corresponds to the skimmer located right before the BSQ (i.e., not the "skimmer 2" component also present in all versions of the Vocus instrument evaluated here.)"

**Comment 6**: Line 379: Based on Fig. S2, the ratio of F1/parent ion changes between ~1 to ~1.75 depending on deltaV1. This seems like a significant difference. The way the PIDs are shown may dampen the impression of how strongly parent ion to fragment ratios change between Vocus settings. Notably, the ratio is what is used to correct ambient data later in the manuscript.

We thank the reviewer for highlighting this. This comment gets to a similar question posed by Reviewer 1 in comment 3 and we point the reviewer to the response of that comment. We have addressed this with updated measurements and a corresponding presentation of data and discussions in Section 3.1.3 Influence of Ion Optic Voltages and Capillary Distance on PIDs of the main text.

**Comment 7**: Line 387: I think the statement here (regarding ion optics) is not supported by enough evidence as it is based on just one VOC and one instrument. With the possibility of some Vocus users not using the Brophy & Farmer method, it has been shown that you can have intense fragmentation on ion optics voltages alone for certain functional groups (Coggon et al., 2024, Fig. S2).

True. We have now addressed this point with updated measurements and a corresponding presentation of data and discussions in Section 3.1.3 Influence of Ion Optic Voltages and Capillary Distance on PIDs of the main text.

**Comment 8**: Figure 3: Which BSQ voltage was used here? It would be helpful to indicate in the caption.

A BSQ voltage of 300 V was used here. We have added that information to the caption of Figure 3.

**Comment 9**: Line 326: Possible typo, change 'the contributions fragment ions in the mass spectrum.' to 'the contributions *of* fragment ions in the mass spectrum.'

Edited, thank you!

**Comment 10**: Line 332 ff: Is it possible that due to the close proximity of the sample inlet capillary to the ion source, the sample gas containing O2 and N2 also enters the ionization region through backdrift and causes impurities independent of a clean water supply?

As the reviewer notes in a later comment, our comparison of the instruments that use higher flowrates compared to the lower flowrate instruments provides some evidence to suggest air

mixing is a possible mechanism for controlling the generation of $NO^+$ and $O_2^+$ reagent ions. We have included the hypothesis in the manuscript that mixing of sample air may dilute the water vapor saturated air in the ionization region thus generating more $NO^+$ and $O_2^+$ reagent ions.

"The increased inlet flowrate may increase mixing of sample air and dilute the water vapor saturated air in the ionization region thus generating more $NO^+$ and $O_2^+$ reagent ions."

**Comment 11**: Line 363: The BSQ is described as impacting ion transmission to the mass spectrometer and its use as a high-pass mass filter is noted in the text. How is it possible that an ion filter is causing ions with m/z 121.09 and above to be so greatly impacted by the BSQ voltage increasing (i.e, the blue line in figure 4a and 4c)? High BSQ voltages are described as causing more filtering, however, there is much higher ion signal from the water clusters at high BSQ voltages. Your results make it seem as though the changing BSQ is changing the ion chemistry in addition to just filtering. In figure 4c, the black line should also be relatively flat, but there is a pronounced 'dip' in the signal from BSQ 300 V to 400 V. This is unexpected for a pure mass filter. Do the authors suspect the BSQ is impacting ion chemistry or fragmentation in addition to serving as the high pass mass filter? Is it possible the pressures in the BSQ are high enough to allow collisions? The inconsistency between expected BSQ trends and what is observed should be discussed.

We re-did these measurements at an E/N of 133 Td and observed a similar pattern in the $MH^+$/water cluster ion formation as the previous measurements indicate. We decided to replace the old figure with a new figure showing only the results measured at E/n = 133 Td because it shows less data while still demonstrating that the BSQ can impact PIDs.

As the reviewer notes, the most peculiar aspect of these data is the $MH^+$ absolute signal decreasing sharply approximately at a BSQ voltage = 300 V. We would have also expected a flat line for an ideal high-pass filter because the $MH^+$ ion has an m/q that should be above the high-pass cutoff at all BSQ voltages. We do not have a good explanation for this, but there are three things to possibly consider to place this observation into context: (1) Holzinger, et al. (2018) models transmission through the Vocus (and other PTR-MS instruments) as a combination of both low-pass and high-pass filters, suggesting there is some aspect of the ion transmission in the Vocus, aside from the ToF duty cycle, that can cause attenuated transmission of higher m/q ions, (2) most transmission curves published in the literature that only assume the BSQ acts as a high-pass filter measure a relative transmission efficiency of $D_5$ siloxane less than 1, and (3) despite a two order of magnitude increase in the mean free path in the BSQ compared to the IMR ion collisions could still occur thus affecting PIDs.

We included a version of the figure that is in the preprint below (for review purposes only) that also includes total ion signal as a function of BSQ voltage. The drop in $MH^+$ signal may be associated with a drop in total ion transmission between BSQ voltage = 225 V and 300 V (panel c below). At 225 V the total ion signal is $5 \times 10^6$ ions s$^{-1}$ and at 300 V the ion signal is $5 \times 10^5$ ions s$^{-1}$ (higher voltages show lower TIC values but remain above $1 \times 10^5$ ions s$^{-1}$ at 450 V). Most of this TIC decrease is likely associated with decreased reagent ion transmission, but none the less demonstrates a large drop in TIC. We hypothesize a drop in total ion transmission could be responsible for both the absolute and relative changes in the $MH^+$ absolute signal and

contribution to the PID, but this is too speculative to include in the text. Constraining ion transmission in PTR-MS warrants a study of its own.

We added text to the manuscript:

"Notably, we cannot explain why the integrated ion counts for the MH$^+$ ion from pentanoic acid decrease going from a BSQ voltage of 200 V to 300 V."

[Figure]

**Comment 12**: Line 387: possibly note that capillary distance can change sensitivity? (if this was observed in your study)

We chose not to go into sensitivity effects of these tests in this manuscript to simplify the message about PIDs. The goal of future work is to relate changes in PIDs to instrument effects on sensitivity/limits of detection.

**Comment 13**: Line 419: Here, acetone is described as challenging to generalize. Later, it is used as the principal example for applying PID results to field data (e.g., line 687 says the method produces reasonable results).

We agree this is confusing. We have added text to clarify the challenge of making general statements about the differences in instrument configuration and trying to predict how they might be responsible for producing the measured PIDs in the interlaboratory comparison versus using a PID from a single instrument to perform quantification.

(Line 496) "We note that the effects of instrument configuration (i.e., E/N, BSQ voltage, ion optic voltages) should have predictable effects on PIDs measured by a single instrument and thus using the product ion quantification methods described later in Section 3.5 are not dependent on our ability to reconcile instrument-to-instrument differences."

**Comment 14**: Line 449 ff: The conclusion that higher $O_2^+$ and $NO^+$ impurities are related to a higher inlet flow should be discussed further in terms of what this means physically. I think this shows that there is substantial drift of sample flow into the ionization region in the Vocus source.

Thank you, this is a good suggestion. We have included the hypothesis in the manuscript that mixing of sample air may dilute the water vapor saturated air in the ionization region thus generating more $NO^+$ and $O_2^+$ reagent ions.

"The increased inlet flowrate may increase mixing of sample air and dilute the water vapor saturated air in the ionization region thus generating more $NO^+$ and $O_2^+$ reagent ions."

**Comment 15**: Line 501 ff: Do the authors think that potentially another factor that influences the PIDs over time within one instrument could be ion source degradation/dirtiness? A dirty ion source could be related to a changing ion source voltage that may impact ion distributions. Were there any tests done to check the impact of the ion source state?

This is a good point. We did not ask anyone to report how dirty the ion source of their instrument was when providing data. The only metric we considered to reflect how dirty the source may be is the ion source voltage. All of the instruments had ion source voltages between 420 V and 440 V with no clear relationship between ion source voltage and fragmentation/clustering/secondary ion chemistry effects on PIDs.

We have added an additional column to the H3O+ PID Library "Lab_ID" tab listing the ion source voltages for each instrument that provided measurements.

**Comment 16**: Line 520: for monoterpenes, there are more papers reporting product ion distributions that may be relevant here:

- Kari, E., Miettinen, P., Yli-Pirilä, P., Virtanen, A., and Faiola, C. L.: PTR-ToF-MS product ion distributions and humidity-dependence of biogenic volatile organic compounds, International Journal of Mass Spectrometry, 430, 87–97,

https://doi.org/10.1016/j.ijms.2018.05.003, available at:
http://www.sciencedirect.com/science/article/pii/S1387380617304943, 2018.

- Tani, A.: Fragmentation and Reaction Rate Constants of Terpenoids Determined by Proton Transfer Reaction-mass Spectrometry, Environmental Control in Biology, 51, 23–29, https://doi.org/10.2525/ecb.51.23, 2013.

Thank you, we have included these references in our discussion of monoterpene product ion interferences.

**Comment 17**: Line 554: 6-MHO is usually 6-methyl-5-hepten-2-one (not 6-methyl-5-heptan-2-one as mentioned in the manuscript) when discussing skin oil oxidation products. Was the 6-MHO used in this study a different molecule or is this just a typo? If you did use a saturated ketone here, it should be mentioned explicitly that this isn't the 6-MHO typically referred to in indoor air literature. If it is the typical 6-MHO, where 6-MHO and 2-octanone are directly compared, it should be noted that one is an unsaturated ketone which could have different ion chemistry available.

Thank you for catching this. This was indeed a typo. We have text in the manuscript that addresses the functional group differences between 2-octanone and 6-MHO below:

"Additionally, as demonstrated by the PID from 6-MHO, adding carbon branching and/or additional functionalities can change product ion formation considerably compared to 2-octanone—the saturated $C_8$ ketone analogue."

**Comment 18**: Figure 10: Consider changing the blue label in panel (b) to "acetone water cluster (calculated)" or something similar to increase clarity.

 We have edited the figure.

**Comment 19**: Line 798: An email address does not seem ideal for keeping up a library that is supposed to be accessible and added to in the future. Are there any plans to set it up as a website with a permanent DOI and a contact button that will be available even if at some point the email address is no longer active?

We have edited the text to reflect the webpage/DOI created for the PID library.

**"Supplement**

Additional analyses of instrument configuration on PIDs are presented in the supplement. A spreadsheet containing the PID data from the interlaboratory comparison (the "$H_3O^+$ PID Library") is included as a supplemental document and the most up-to-date versions can be retrieved online (doi:10.18434/mds2-3582). Users wishing to submit data to this library can email the corresponding author (michael.f.link@nist.gov) and a link to submit a data file will be

provided. More details can be found in the "ReadMe" tab of the supplemental $H_3O^+$ PID Library."